# What Makes Multi-modal Learning Better than Single (Provably)

**Yu Huang**[1,*]**, Chenzhuang Du**[1,*]**, Zihui Xue**[2]**, Xuanyao Chen**[3,4]**,**

**Hang Zhao**[1,4]**, Longbo Huang**[1,†]

[1] Institute for Interdisciplinary Information Sciences, Tsinghua University
[2] The University of Texas at Austin   [3] Fudan University
[4] Shanghai Qi Zhi Institute

## Abstract

The world provides us with data of multiple modalities. Intuitively, models fusing data from different modalities outperform their uni-modal counterparts, since more information is aggregated. Recently, joining the success of deep learning, there is an influential line of work on deep multi-modal learning, which has remarkable empirical results on various applications. However, theoretical justifications in this field are notably lacking.

*Can multi-modal learning provably perform better than uni-modal?*

In this paper, we answer this question under a most popular multi-modal fusion framework, which firstly encodes features from different modalities into a common latent space and seamlessly maps the latent representations into the task space. We prove that learning with multiple modalities achieves a smaller population risk than only using its subset of modalities. The main intuition is that the former has a more accurate estimate of the latent space representation. To the best of our knowledge, this is the first theoretical treatment to capture important qualitative phenomena observed in real multi-modal applications from the generalization perspective. Combining with experiment results, we show that multi-modal learning does possess an appealing formal guarantee.

## 1   Introduction

Our perception of the world is based on different modalities, *e.g.* sight, sound, movement, touch, and even smell [36]. Inspired from the success of deep learning [26, 21], deep multi-modal research is also activated, which covers fields like audio-visual learning [8, 43], RGB-D semantic segmentation [35, 20] and Visual Question Answering [17, 2].

While deep multi-modal learning shows excellent power in practice, theoretical understanding of deep multi-modal learning is limited. Some recent works have been done towards this direction [39, 48]. However, these works made strict assumptions on the probability distributions across different modalities, which may not hold in real-life applications [30]. Notably, they do not take *generalization* performance of multi-modal learning into consideration. Toward this end, the following fundamental problem remains largely open:

---

*equal contribution
†Correspondence to: longbohuang@tsinghua.edu.cn

35th Conference on Neural Information Processing Systems (NeurIPS 2021).

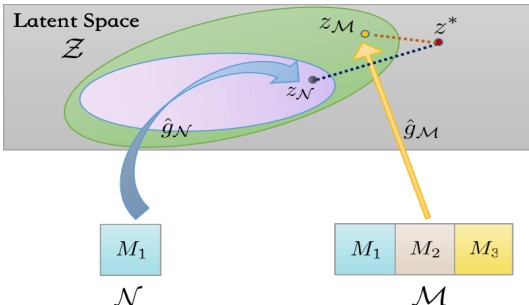

Figure 1: $\mathcal{M}$ *vs* $\mathcal{N}$ **modalities latent space representation**, where the latter is a subset of the former. $z_{\mathcal{M}}$, $z_{\mathcal{N}}$ and $z^{\star}$ are images on the latent space $\mathcal{Z}$ corresponding to the representation mappings $\hat{g}_{\mathcal{M}}$, $\hat{g}_{\mathcal{N}}$ and $g^{\star}$. $M_i$ denotes modality $i$.

*Can multi-modal learning provably performs better than uni-modal?*

In this paper, we provably answer this question from two perspectives:

- (When) Under what conditions multi-modal performs better than uni-modal?
- (Why) What results in the performance gains ?

The framework we study is abstracted from the multi-modal fusion approaches, which is one of the most researched topics of multi-modal learning [3]. Specifically, we first encode the complex data from heterogeneous sources into a common latent space $\mathcal{Z}$. The true latent representation is $g^{\star}$ in a function class $\mathcal{G}$, and the task mapping $h^{\star}$ is contained in a function class $\mathcal{H}$ defined on the latent space. Our model corresponds to the recent progress of deep multi-modal learning on various applications, such as scene classification [9] and action recognition [24, 43].

Under this composite framework, we provide the first theoretical analysis to shed light on what makes multi-modal outperform uni-modal from the generalization perspective. We identify the relationship between the population risk and the distance between a learned latent representation $\hat{g}$ and the $g^{\star}$, under the metric we will define later. Informally, closer to the true representation leads to less population loss, which indicates that a better latent representation guarantees the end-to-end multi-modal learning performance. Instead of simply considering the comparison of *multi vs uni* modalities, we consider a general case, $\mathcal{M}$ *vs* $\mathcal{N}$ modalities, which are distinct subsets of all modalities. We focus on the condition that the latter is a subset of the former. Our second result is a bound for the closeness between $\hat{g}$ and the $g^{\star}$, from which we provably show that the latent representation $\hat{g}_{\mathcal{M}}$ learning from the $\mathcal{M}$ modalities is closer to the true $g^{\star}$ than $\hat{g}_{\mathcal{N}}$ learning from $\mathcal{N}$ modalities. As shown in Figure 1, $\hat{g}_{\mathcal{M}}$ has a more sufficient latent space exploration than $\hat{g}_{\mathcal{N}}$. Moreover, in a specific linear regression model, we directly verify that using multiple modalities rather than its subset learns a better latent representation.

The main contributions of this paper are summarized as follows:

- We formalize the multi-modal learning problem into a theoretical framework. Firstly, we show that the performance of multi-modal learning in terms of population risk can be bounded by the *latent representation quality*, a novel metric we propose to measure the distance from a learned latent representation to the true representation, which reveals that the ability of learning the whole task coincides with the ability of learning the latent representation when we have sufficient training samples.

- We derive an upper bound for the latent representation quality of training over a subset of modalities. This directly implies a principle to guide us in modality selection, i.e., when the number of sample size is large and multiple modalities can efficiently optimize the empirical risk, using multi-modal to build a recognition or detection system can have a better performance.

- Restricted to linear latent and task mapping, we provide rigorous theoretical analysis that latent representation quality degrades when the subset of multiple modalities is applied.

Experiments are also carried out to empirically validate the theoretical observation that $\hat{g}_{\mathcal{N}}$ is inferior to $\hat{g}_{\mathcal{M}}$.

The rest of the paper is organized as follows. In the next section, we review the related literature. The formulation of multi-modal learning problem is described in Section 3. Main results are presented in Section 4. In Section 5, we show simulation results to support our theoretical claims. Additional discussions about the inefficiency of multi-modal learning are presented in Section 6. Finally, conclusions are drawn in Section 7.

## 2 Related Work

**Multi-modal Learning Applications**  Deep learning makes fusing different signals easier, which enables us to develop many multi-modal frameworks. For example, [35, 23, 22, 31, 20] combine RGB and depth images to improve semantic segmentation; [8, 16] fuse audio with video to do scene understanding; researchers also explore audio-visual source separation and localization [49, 11].

**Theory of Multi-modal Learning**  On the theory side, in semi-supervised setting, [39] proposed a novel method, Total Correlation Gain Maximization (TCGM), and theoretically proved that TCGM can find the groundtruth Bayesian classifier given each modality. Moreover, CPM-Nets [48] showed multi-view representation can recover the same performance as only using the single-view observation by constructing the versatility. However, they made strict assumptions on the relationship across different modalities, while our analysis does not require such additional assumptions. Besides, previous multi-view analysis [37, 1, 45, 13] typically assumes that each view alone is sufficient to predict the target accurately, which may not hold in our multi-modal setting. For instance, it is difficult to build a classifier just using a weak modality with limited labeled data, e.g., depth modality in RGB-D images for object detection task [18].

**Transfer Learning**  A line of work closely related to our composite learning framework is transfer learning via representation learning, which firstly learns a shared representation on various tasks and then transfers the learned representation to a new task. [40, 7, 32, 10] have provided the sample complexity bounds in the special case of the linear feature and linear task mapping. [41] introduces a new notion of task diversity and provides a generalization bound with general tasks, features, and losses. Unfortunately, the function class which contains feature mappings is the same across all different tasks while our focus is that the function classes generated by different subsets modalities are usually inconsistent.

**Notation:**  Throughout the paper, we use $\|\cdot\|$ to denote the $\ell_2$ norm. We also denote the set of positive integer numbers less or equal than $n$ by $[n]$, i.e. $[n] \triangleq \{1, 2, \cdots, n\}$.

## 3 The Multi-modal Learning Formulation

In this section, we present the Multi-modal Learning problem formulation. Specifically, we assume that a given data $\mathbf{x} := \left(x^{(1)}, \cdots, x^{(K)}\right)$ consists of $K$ modalities, where $x^{(k)} \in \mathcal{X}^{(k)}$ the domain set of the $k$-th modality. Denote $\mathcal{X} = \mathcal{X}^{(1)} \times \cdots \times \mathcal{X}^{(K)}$. We use $\mathcal{Y}$ to denote the target domain and use $\mathcal{Z}$ to denote a latent space. Then, we denote $g^\star : \mathcal{X} \mapsto \mathcal{Z}$ the true mapping from the input space (using all of $K$ modalities) to the latent space, and $h^\star : \mathcal{Z} \mapsto \mathcal{Y}$ is the true task mapping. For instance, in aggregation-based multi-modal fusion, $g^\star$ is an aggregation function compounding on $K$ seperate sub-networks and $h^\star$ is a multi-layer neural network [44].

In the learning task, a data pair $(\mathbf{x}, y) \in \mathcal{X} \times \mathcal{Y}$ is generated from an unknown distribution $\mathcal{D}$, such that

$$\mathbb{P}_{\mathcal{D}}(\mathbf{x}, y) \triangleq \mathbb{P}_{y|\mathbf{x}}\left(y \mid h^\star \circ g^\star(\mathbf{x})\right) \mathbb{P}_{\mathbf{x}}(\mathbf{x}) \tag{1}$$

Here $h^\star \circ g^\star(\mathbf{x}) = h^\star(g^\star(\mathbf{x}))$ represents the composite function of $h^\star$ and $g^\star$.

In real-world settings, we often face incomplete multi-modal data, i.e., some modalities are not observed. To take into account this situation, we let $\mathcal{M}$ be a subset of $[K]$, and without loss of generality, focus on the learning problem only using the modalities in $\mathcal{M}$. Specifically, define

$\mathcal{X}' := \left(\mathcal{X}^{(1)} \cup \{\bot\}\right) \times \ldots \times \left(\mathcal{X}^{(K)} \cup \{\bot\}\right)$ as the extension of $\mathcal{X}$, where $\mathbf{x}' \in \mathcal{X}'$, $\mathbf{x}'_k = \bot$ means that the $k$-th modality is not used (collected). Then we define a mapping $p_\mathcal{M}$ from $\mathcal{X}$ to $\mathcal{X}'$ induced by $\mathcal{M}$:

$$p_\mathcal{M}(\mathbf{x})^{(k)} = \begin{cases} \mathbf{x}^{(k)} & \text{if } k \in \mathcal{M} \\ \bot & \text{else} \end{cases}$$

Also define $p'_\mathcal{M} : \mathcal{X}' \mapsto \mathcal{X}'$ as the extension of $p_\mathcal{M}$. Let $\mathcal{G}'$ denote a function class, which contains the mapping from $\mathcal{X}'$ to the latent space $\mathcal{Z}$, and define a function class $\mathcal{G}_\mathcal{M}$ as follows:

$$\mathcal{G}_\mathcal{M} \triangleq \{g_\mathcal{M} : \mathcal{X} \mapsto \mathcal{Z} \mid g_\mathcal{M}(\mathbf{x}) := g'(p_\mathcal{M}(\mathbf{x})), g' \in \mathcal{G}'\} \tag{2}$$

Given a data set $\mathcal{S} = ((\mathbf{x}_i, y_i))_{i=1}^m$, where $(\mathbf{x}_i, y_i)$ is drawn i.i.d. from $\mathcal{D}$, the learning objective is, following the Empirical Risk Minimization (ERM) principle [27], to find $h \in \mathcal{H}$ and $g_\mathcal{M} \in \mathcal{G}_\mathcal{M}$ to jointly minimize the empirical risk, i.e.,

$$\min \quad \hat{r}\left(h \circ g_\mathcal{M}\right) \triangleq \frac{1}{m} \sum_{i=1}^m \ell\left(h \circ g_\mathcal{M}(\mathbf{x}_i), y_i\right) \tag{3}$$

$$\text{s.t.} \quad h \in \mathcal{H}, g_\mathcal{M} \in \mathcal{G}_\mathcal{M}. \tag{4}$$

where $\ell(\cdot, \cdot)$ is the loss function. Given $\hat{r}\left(h \circ g_\mathcal{M}\right)$, we similarly define its corresponding population risk as

$$r\left(h \circ g_\mathcal{M}\right) = \mathbb{E}_{(\mathbf{x}_i, y_i) \sim \mathcal{D}}\left[\hat{r}\left(h \circ g_\mathcal{M}\right)\right] \tag{5}$$

Similar to [1, 41], we use the population risk to measure the performance of learning.

**Example.** As a concrete example of our model, consider the video classification problem under the late-fusion model in [43]. In this case, each modality $k$, e.g. RGB frames, audio or optical flows, is encoded by a deep network $\varphi_k$, and their features are fused and passed to a classifier $\mathcal{C}$. If we train on the first $M$ modalities, we can let $\mathcal{M} = [M]$. Then $g_\mathcal{M}$ has the form: $\varphi_1 \oplus \varphi_2 \oplus \cdots \oplus \varphi_M$, where $\oplus$ denotes a fusion operation, e.g. self-attention ($\mathcal{Z}$ is the output of $g_\mathcal{M}$), and $h$ is the classifier $\mathcal{C}$. More examples are provided in Appendix B.

**Why Composite Framework ?** Note that the composite multi-modal framework is often observed in applications. In fact, in recent years, a large number of papers, e.g., [3, 12, 14, 44, 43, 24], appear to have utilized this framework in one way or another, even though the contributors did not clearly summarize the relationship between their methods and this common underlying structure. However, despite the popularity of the framework, existing works lack a very formal definition in theory.

**What is Special about Multi-modal Learning?** For the multi-modal representation $\mathbf{x} := \left(x^{(1)}, \cdots, x^{(K)}\right)$ consists of $K$ modalities, we allow the dimension of the domain set of each modality $\mathcal{X}^{(k)}$ to be different, which well models the source of heterogeneity of each modality. The relationships across different modalities are usually of varying levels due to the heterogeneous sources. Therefore, compared to previous works [45, 48], we make no assumptions on the relationship across every single modality in our analysis, which makes it general to allow different correlations. Moreover, the main assumption behind previous analysis [37, 39, 42] is that each view/modality contains sufficient information for target tasks, which does not shed light on our analysis. It may not hold in multi-modal applications [47], e.g., in object detecting task, it is known that depth images are with lower accuracy than RGB images [18].

## 4 Main Results

In this section, we provide main theoretical results to rigorously establish various aspects of the folklore claim that multi-modal is better than single. We first detail several assumptions throughout this section.

**Assumption 1.** *The loss function $\ell(\cdot, \cdot)$ is L-smooth with respect to the first coordinate, and is bounded by a constant $C$.*

**Assumption 2.** *The true latent representation $g^\star$ is contained in $\mathcal{G}$, and the task mapping $h^\star$ is contained in $\mathcal{H}$.*

Assumption 1 is a classical regularity condition for loss function in theoretical analysis [27, 41, 40]. Assumption 2 is also be known as realizability condition in representation learning [41, 10, 40], which ensures that the function class that we optimize over contains the true latent representation and the task mapping.

**Assumption 3.** *For any $g' \in \mathcal{G}'$ and $\mathcal{M} \subset [K]$, $g' \circ p'_{\mathcal{M}} \in \mathcal{G}'$.*

To understand Assumption 3, note that for any $\mathcal{N} \subset \mathcal{M} \subset [K]$, by definition, for any $g_{\mathcal{N}} \in \mathcal{G}_{\mathcal{N}}$, there exists $g' \in \mathcal{G}'$, s.t.
$$g_{\mathcal{N}}(\mathbf{x}) = g'(p_{\mathcal{N}}(\mathbf{x})) = g'(p'_{\mathcal{N}}(p_{\mathcal{M}}(\mathbf{x})))$$
Therefore, Assumption 3 directly implies $g_{\mathcal{N}} \in \mathcal{G}_{\mathcal{M}}$. Moreover, we have $\mathcal{G}_{\mathcal{N}} \subset \mathcal{G}_{\mathcal{M}} \subset \mathcal{G}$, which means that the inclusion relationship of modality subsets remains unchanged on the latent function class induced by them. As an example, if $\mathcal{G}'$ is linear, represented as matrix $\mathbf{G} \in \mathbb{R}^{Q \times K}$. Also $p'_{\mathcal{M}}$ can be represented as a diagonal matrix $\mathbf{P} \in \mathbb{R}^{K \times K}$ with the $i$-th diagonal entry being 1 for $i \in \mathcal{M}$ and 0 otherwise. In this case, Assumption 3 holds, i.e. $\mathbf{G} \times \mathbf{P} \in \mathcal{G}'$. Moreover, $\mathbf{G} \times \mathbf{P}$ is a matrix with $i$-th column all be zero for $i \notin \mathcal{M}$, which is commonly used in the underfitting analysis in linear regression [34].

### 4.1 Connection to Latent Representation Quality

Latent space is employed to better exploit the correlation among different modalities. Therefore, we will naturally conjecture that the performance of training with different modalities is related to its ability to learn latent space representation. In this section, we will formally characterize this relationship.

In order to measure the goodness of a learned latent representation $g$, we introduce the following definition of *latent representation quality*.

**Definition 1.** Given a data distribution with the form in (1), for any learned latent representation mapping $g \in \mathcal{G}$, the **latent representation quality** is defined as
$$\eta(g) = \inf_{h \in \mathcal{H}} \left[ r\left(h \circ g\right) - r(h^* \circ g^*) \right] \tag{6}$$

Here $\inf_{h \in \mathcal{H}} r\left(h \circ g\right)$ is the best achievable population risk with the fixed latent representation $g$. Thus, to a certain extent, $\eta(g)$ measures the loss incurred by the distance between $g$ and $g^{\star}$.

Next, we recap the Rademacher complexity measure for model complexity. It will be used in quantifying the population risk performance based on different modalities. Specifically, let $\mathcal{F}$ be a class of vector-valued function $\mathbb{R}^d \mapsto \mathbb{R}^n$. Let $Z_1, \ldots, Z_m$ be i.i.d. random variables on $\mathbb{R}^d$ following some distribution $P$. Denote the sample $S = (Z_1, \ldots, Z_m)$. The empirical Rademacher complexity of $\mathcal{F}$ with respect to the sample $S$ is given by [5]
$$\widehat{\mathfrak{R}}_S(\mathcal{F}) := \mathbb{E}_{\sigma} \left[ \sup_{f \in \mathcal{F}} \frac{1}{m} \sum_{i=1}^{m} \sigma_i f\left(Z_i\right) \right]$$
where $\sigma = (\sigma_1, \ldots, \sigma_n)^{\top}$ with $\sigma_i \overset{iid}{\sim} \text{unif}\{-1, 1\}$. The Rademacher complexity of $\mathcal{F}$ is
$$\mathfrak{R}_m(\mathcal{F}) = \mathbb{E}_S \left[ \widehat{\mathfrak{R}}_S(\mathcal{F}) \right]$$

Now we present our first main result regarding multi-modal learning.

**Theorem 1.** *Let $\mathcal{S} = ((\mathbf{x}_i, y_i))_{i=1}^{m}$ be a dataset of $m$ examples drawn i.i.d. according to $\mathcal{D}$. Let $\mathcal{M}, \mathcal{N}$ be two distinct subsets of $[K]$. Assuming we have produced the empirical risk minimizers $(\hat{h}_{\mathcal{M}}, \hat{g}_{\mathcal{M}})$ and $(\hat{h}_{\mathcal{N}}, \hat{g}_{\mathcal{N}})$, training with the $\mathcal{M}$ and $\mathcal{N}$ modalities separately. Then, for all $1 > \delta > 0$, with probability at least $1 - \frac{\delta}{2}$:*
$$r\left(\hat{h}_{\mathcal{M}} \circ \hat{g}_{\mathcal{M}}\right) - r\left(\hat{h}_{\mathcal{N}} \circ \hat{g}_{\mathcal{N}}\right)$$
$$\leq \gamma_{\mathcal{S}}(\mathcal{M}, \mathcal{N}) + 8L\mathfrak{R}_m(\mathcal{H} \circ \mathcal{G}_{\mathcal{M}}) + \frac{4C}{\sqrt{m}} + 2C\sqrt{\frac{2\ln(2/\delta)}{m}} \tag{7}$$
*where*
$$\gamma_{\mathcal{S}}(\mathcal{M}, \mathcal{N}) \triangleq \eta(\hat{g}_{\mathcal{M}}) - \eta(\hat{g}_{\mathcal{N}}) \qquad \square \tag{8}$$

**Remark.** A few remarks are in place. First of all, $\gamma_{\mathcal{S}}(\mathcal{M}, \mathcal{N})$ defined in (8) compares the quality between latent representations learning from $\mathcal{M}$ and $\mathcal{N}$ modalities with respect to the given dataset $\mathcal{S}$. Theorem 1 bounds the difference of population risk training with two different subsets of modalities by $\gamma_{\mathcal{S}}(\mathcal{M}, \mathcal{N})$, which validates our conjecture that including more modalities is advantageous in learning. Second, for the commonly used function classes in the field of machine learning, Radamacher complexity for a sample of size $m$, $\mathfrak{R}_m(\mathcal{F})$ is usually bounded by $\sqrt{C(\mathcal{F})/m}$, where $C(\mathcal{F})$ represents the intrinsic property of function class $\mathcal{F}$. Third, (7) can be written as $\gamma_{\mathcal{S}}(\mathcal{M}, \mathcal{N}) + \mathcal{O}(\sqrt{\frac{1}{m}})$ in order terms. This shows that as the number of sample size grows, the performance of using different modalities mainly depends on its latent representation quality.

## 4.2 Upper Bound for Latent Space Exploration

Having establish the connection between the population risk difference with latent representation quality, our next goal is to estimate how close the learned latent representation $\hat{g}_{\mathcal{M}}$ is to the true latent representation $g^{\star}$. The following theorem shows how the latent representation quality can be controlled in the training process.

**Theorem 2.** *Let* $\mathcal{S} = \{(\mathbf{x}_i, y_i)\}_{i=1}^m$ *be a dataset of* $m$ *examples drawn i.i.d. according to* $\mathcal{D}$. *Let* $\mathcal{M}$ *be a subset of* $[K]$. *Assuming we have produced the empirical risk minimizers* $(\hat{h}_{\mathcal{M}}, \hat{g}_{\mathcal{M}})$ *training with the* $\mathcal{M}$ *modalities. Then, for all* $1 > \delta > 0$, *with probability at least* $1 - \delta$:

$$\eta(\hat{g}_{\mathcal{M}}) \leq 4L\mathfrak{R}_m(\mathcal{H} \circ \mathcal{G}_{\mathcal{M}}) + 4L\mathfrak{R}_m(\mathcal{H} \circ \mathcal{G}) + 6C\sqrt{\frac{2\ln(2/\delta)}{m}} + \hat{L}(\hat{h}_{\mathcal{M}} \circ \hat{g}_{\mathcal{M}}, \mathcal{S}) \qquad (9)$$

*where* $\hat{L}(\hat{h}_{\mathcal{M}} \circ \hat{g}_{\mathcal{M}}, \mathcal{S}) \triangleq \hat{r}\left(\hat{h}_{\mathcal{M}} \circ \hat{g}_{\mathcal{M}}\right) - \hat{r}\left(h^{\star} \circ g^{\star}\right)$ *is the centered empirical loss.* $\square$

**Remark.** Consider sets $\mathcal{N} \subset \mathcal{M} \subset [K]$. Under Assumption 3, $\mathcal{G}_{\mathcal{N}} \subset \mathcal{G}_{\mathcal{M}} \subset \mathcal{G}$, optimizing over a larger function class results in a smaller empirical risk. Therefore

$$\hat{L}(\hat{h}_{\mathcal{M}} \circ \hat{g}_{\mathcal{M}}, \mathcal{S}) \leq \hat{L}(\hat{h}_{\mathcal{N}} \circ \hat{g}_{\mathcal{N}}, \mathcal{S}) \qquad (10)$$

Similar to the analysis in Theorem 1, the first term on the Right-hand Side (RHS), $\mathfrak{R}_m(\mathcal{H} \circ \mathcal{G}_{\mathcal{M}}) \sim \sqrt{C(\mathcal{H} \circ \mathcal{G}_{\mathcal{M}})/m}$ and $\mathfrak{R}_m(\mathcal{H} \circ \mathcal{G}_{\mathcal{N}}) \sim \sqrt{C(\mathcal{H} \circ \mathcal{G}_{\mathcal{N}})/m}$. Following the basic structural property of Radamacher complexity [5], we have $C(\mathcal{H} \circ \mathcal{G}_{\mathcal{N}}) \leq C(\mathcal{H} \circ \mathcal{G}_{\mathcal{M}})$. Therefore, Theorem 2 offers the following principle for choosing modalities to improve the latent representation quality.

**Principle:** *choose to learn with more modalities if:*

$$\hat{L}(\hat{h}_{\mathcal{N}} \circ \hat{g}_{\mathcal{N}}, \mathcal{S}) - \hat{L}(\hat{h}_{\mathcal{M}} \circ \hat{g}_{\mathcal{M}}, \mathcal{S}) \geq \sqrt{\frac{C(\mathcal{H} \circ \mathcal{G}_{\mathcal{M}})}{m}} - \sqrt{\frac{C(\mathcal{H} \circ \mathcal{G}_{\mathcal{N}})}{m}}$$

What this principle implies are twofold. (i) When the number of sample size $m$ is large, the impact of intrinsic complexity of function classes will be reduced. (ii) Using more modalities can efficiently optimize the empirical risk, hence improve the latent representation quality.

Through the trade-off illustrated in the above principle, we provide theoretical evidence that when $\mathcal{N} \subset \mathcal{M}$ and training samples are sufficient, $\eta(\hat{g}_{\mathcal{M}})$ may be less than $\eta(\hat{g}_{\mathcal{N}})$, i.e. $\gamma_{\mathcal{S}}(\mathcal{M}, \mathcal{N}) \leq 0$. Moreover, combining with the conclusion from Theorem 1, if the sample size $m$ is large enough, $\gamma_{\mathcal{S}}(\mathcal{M}, \mathcal{N}) \leq 0$ guarantees $r\left(\hat{h}_{\mathcal{M}} \circ \hat{g}_{\mathcal{M}}\right) \leq r\left(\hat{h}_{\mathcal{N}} \circ \hat{g}_{\mathcal{N}}\right)$, which indicates learning with the $\mathcal{M}$ modalities outperforms only using its subset $\mathcal{N}$ modalities.

**Role of the intrinsic property** $\mathcal{C}(\cdot)$   Hypothesis function classes are typically overparametrized in deep learning, and will be extremely large for some typical measures, e.g., VC dimension, absolute dimension. Some recent efforts aim to offer an explanation about why neural networks generalize better with over-parametrization [28, 29, 4]. [28] suggest a novel complexity measure based on unit-wise capacities, which implies if both in overparametrized settings, the complexity will not change much or even decrease when we have more modalities (using more parameters). Thus, the inequality in the principle is trivially satisfied, and we will choose to learn with more modalities.

## 4.3 Non-Positivity Guarantee

In this section, we focus on a composite linear data generating model to theoretically verify that the $\gamma_{\mathcal{S}}(\mathcal{M}, \mathcal{N})$ is indeed non-positive in this special case.[3] Specifically, we consider the case where the mapping to the latent space and the task mapping are both linear. Formally, let the function class $\mathcal{G}$ and $\mathcal{H}$ be:

$$\begin{aligned}
\mathcal{G} &= \left\{ g \mid g(\mathbf{x}) = \mathbf{A}^{\top}\mathbf{x}, \mathbf{A} \in \mathbb{R}^{d \times n}, \mathbf{A} \right\} \\
\mathcal{H} &= \left\{ h \mid h(\mathbf{z}) = \boldsymbol{\beta}^{\top}\mathbf{z}, \boldsymbol{\beta} \in \mathbb{R}^{n}, \|\boldsymbol{\beta}\| \leq C_b \right\}
\end{aligned} \tag{11}$$

where $\mathbf{x} = \left( \mathbf{x}^{(1)}, \cdots, \mathbf{x}^{(K)} \right)$ is a $d$-dimensional vector, $\mathbf{x}^{(k)} \in \mathbb{R}^{d_k}$ denotes the feature vector for the $k$-th modality and $\sum_{k=1}^{K} d_k = d$. Here, the distribution $\mathbb{P}_{\mathbf{x}}(\cdot)$ satisfies that its covariance matrix is positive definite. The data is generated by:

$$y = (\boldsymbol{\beta}^{\star})^{\top} \mathbf{A}^{\star\top}\mathbf{x} + \epsilon \tag{12}$$

where r.v. $\epsilon$ is independent of $\mathbf{x}$ and has zero-mean and bounded second moment. Note that in practical multi-modal learning, usually only one layer is linear and the other is a neural network. For instance, [50] employs the linear matrix to project the feature matrix from different modalities into a common latent space for early dementia diagnosis, i.e., $\mathcal{G}$ is linear. Another example is in pedestrian detection [46], where a linear task mapping is adopted, i.e., $\mathcal{H}$ is linear. Thus, our composite linear model can be viewed as an approximation to such popular models, and our results can offer insights into the performance of these models.

We consider a special case that $\mathcal{M} = [K]$ and $\mathcal{N} = [K-1]$. Thus $\mathcal{G}_{\mathcal{M}} = \mathcal{G}$ and we have the following result.

$$\mathcal{G}_{\mathcal{N}} = \left\{ g \mid g(\mathbf{x}) = \left[ \begin{array}{c} \mathbf{A}_{1:\sum_{k=1}^{K-1} d_k} \\ \mathbf{0} \end{array} \right]^{\top} \mathbf{x}, \mathbf{A} \in \mathbb{R}^{d \times n} \text{ with orthonormal columns} \right\} \tag{13}$$

**Proposition 1.** *Consider the dataset $\mathcal{S} = \{(\mathbf{x}_i, y_i)\}_{i=1}^{m}$ generating from the linear model defined in (12) with $\ell_2$ loss. Let $\mathcal{M} = [K]$ and $\mathcal{N} = [K-1]$. Let $\hat{\mathbf{A}}_{\mathcal{M}}$, $\hat{\mathbf{A}}_{\mathcal{N}}$ denote the projection matrix estimated by $\mathcal{M}$, $\mathcal{N}$ modalities. Assume that $\hat{\mathbf{A}}_{\mathcal{M}}$, $\mathbf{A}^{\star}$ has orthonormal columns. If $n = d$, for sufficiently large constant $C_b$, we have:*

$$\gamma_{\mathcal{S}}(\mathcal{M}, \mathcal{N}) \leq 0 \tag{14}$$

In this special case, Proposition 1 directly guarantees that training with incomplete modalities weakens the ability to learn a optimal latent representation. As a result, it also degrades the learning performance.

# 5 Experiment

We conduct experiments to validate our theoretical results. The source of the data we consider is two-fold, multi-modal real-world dataset and well-designed generated dataset.

## 5.1 Real-world dataset

**Dataset.** The natural dataset we use is the Interactive Emotional Dyadic Motion Capture (IEMO-CAP) database, which is an acted multi-modal and multi-speaker database [6]. It contains three modalities, Text, Video and Audio. We follow the data preprocessing method of [33] and obtain 100 dimensions data for audio, 100 dimensions for text, and 500 dimensions for video. There are six labels here, namely, happy, sad, neutral, angry, excited and frustrated. We use 13200 data for training and 3410 for testing.

**Training Setting.** For all experiments on IEMOCAP, we use one linear neural network layer to extract the latent feature, and we set the hidden dimension to be 128. In multi-modal network, different modalities do not share encoders and we concatenate the features first, and then map the

---

[3]Proving that $\gamma_{\mathcal{S}}(\mathcal{M}, \mathcal{N}) \leq 0$ holds in general is open and will be an interesting future work.

feature to the task space. We use Adam [25] as the optimizer and set the learning rate to be $0.01$, with other hyper-parameters default. The batch size is $2048$ for the data. For this classification task, the top-1 accuracy is used for performance measurement. We use naively multi-modal late-fusion training as our framework [43, 9]. In Appendix C, we provide more discussions on stable multi-modal training.

**Connection to the Latent Representation Quality.** The classification accuracy on IEMOCAP, using different combinations of modalities are summarized in Table 1. All learning strategies using multiple modalities outperform the single modal baseline. To validate Theorem 1, we calculate the test accuracy difference between different subsets of modalities using the result in Table 1 and show them in the third column of Table 3.

Moreover, we empirically evaluate the $\eta(\hat{g}_{\mathcal{M}})$ in the following way: freeze the encoder $\hat{g}_M$ obtained through pretraining and then finetune to obtain a better classifier $h$. Having $\eta(\hat{g}_{\mathcal{M}})$ and $\eta(\hat{g}_{\mathcal{N}})$, the values of $\gamma_{\mathcal{S}}(\mathcal{M}, \mathcal{N})$ between different subsets of modalities are also presented in Table 3. Previous discussions on Theorem 1 implies that the population risk difference between $\mathcal{M}$ and $\mathcal{N}$ modalities has the same sign as $\gamma_{\mathcal{S}}(\mathcal{M}, \mathcal{N})$ when the sample size is large enough, and negativity implies performance gains. Since we use accuracy as the measure, on the contrary, positivity indicates a better performance in our settings. As shown in Table 3, when more modalities are added for learning, the test accuracy difference and $\gamma_{\mathcal{S}}(\mathcal{M}, \mathcal{N})$ are both positive, which confirms the important role of latent representation quality characterized in Theorem 1.

Table 1: Test classification accuracy on IEMOCAP, using different combinations of modalities, only Text, Text + Video, Text + Audio and Text + Video + Audio.

| Modalities | Test Acc |
|---|---|
| Text(T) | 49.93±0.57 |
| Text + Video(TV) | 51.08±0.66 |
| Text + Audio(TA) | 53.03±0.21 |
| Text + Video + Audio(TVA) | **53.89 ± 0.47** |

Table 2: Latent representation quality vs. The number of the sample size on IEMOCAP. Noting that in this table, we show the results from naively end-to-end late-fusion training and in Appendix C, we discuss on more stable multi-modal training methods.

| Modalities | Test Acc (Ratio of Sample Size) | | | | |
|---|---|---|---|---|---|
| | $10^{-4}$ | $10^{-3}$ | $10^{-2}$ | $10^{-1}$ | 1 |
| T | 23.66±1.28 | 29.08±3.34 | 45.63±0.29 | 48.30±1.31 | 49.93±0.57 |
| TA | **25.06±1.05** | 34.28±4.54 | **47.28±1.24** | 50.46±0.61 | 51.08±0.66 |
| TV | 24.71±0.87 | **38.37±3.12** | 46.54±1.62 | 49.50±1.04 | 53.03±0.21 |
| TVA | 24.71±0.76 | 32.24±1.17 | 46.39±3.82 | **50.75±1.45** | **53.89±0.47** |

Table 3: Comparison of test accuracy and latent representation quality among different combinations of modalities.

| $\mathcal{M}$ Modalities | $\mathcal{N}$ Modalities | Test Acc **Difference** | $\gamma_{\mathcal{S}}(\mathcal{M}, \mathcal{N})$ |
|---|---|---|---|
| TA | T | 1.15 | 1.36 |
| TV | T | 3.10 | 3.57 |
| TVA | TV | 0.86 | 0.19 |
| TVA | TA | 2.81 | 2.4 |

**Upper Bound for Latent Space Exploration.** Table 3 also confirms our theoretical analysis in Theorem 2. In all cases, the $\mathcal{N}$ modalities is a subset of $\mathcal{M}$ modalities, and correspondingly, a positive $\gamma_{\mathcal{S}}(\mathcal{M}, \mathcal{N})$ is observed. This indicates that $\mathcal{M}$ modalities has a more sufficient latent space exploration than its subset $\mathcal{N}$ modalities.

We also attempt to understand the use of sample size for exploring the latent space. Table 2 presents the latent representation quality $\eta$ obtained by using different numbers of sample size, which is measured by the test accuracy of the pretrain+finetuned modal. Here, the ratio of sample size is set to the total number of training samples. The corresponding curve is also ploted in Figure 2(a). As the number of sample size grows, the increase in performance of $\eta$ is observed, which is in keeping with the $\mathcal{O}(\sqrt{1/m})$ term in our upper bound for $\eta$. The phenomenon that the combination of Text, Video and Audio (TVA) modalities underperforms the uni-modal when the number of sample size is relatively small, can also be interpreted by the trade-off we discussed in Theorem 2. When there are insufficient training examples, the intrinsic complexity of the function class induced by multiple modalities dominates, thus weakening its latent representation quality.

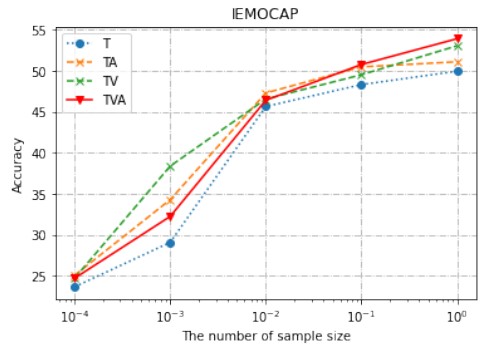

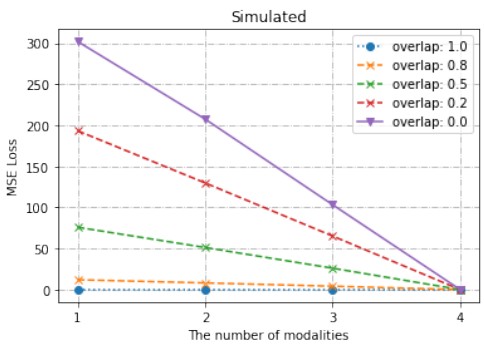

(a) Latent representation quality vs.The ratio of the sample size on IEMOCAP

(b) Latent representation quality vs. The number of modalities on simulated data

Figure 2: Evaluation of the latent representation quality on different data sets

## 5.2 Synthetic Data

In this subsection, we investigate the effect of modality correlation on latent representation quality. Typically, there are three situations for the correlation across each modality [39]. (i) Each modality does not share information at all, that is, each modality only contains modal-specific information. (ii) The other is that, all modalities only maintain the share information without unique information on their own. (iii) The last is a moderate condition, i.e., each modal not only shares information, but also owns modal-specific information. The reason to utilize simulated data in this section is due to the fact that it is hard in practice to have natural datasets that possess the required degree of modality correlation.

**Data Generation.** For the synthetic data, we have four modalities, denoted by $m_1, m_2, m_3, m_4$, respectively, and the generation process is summarized as follows:

Step 1: Generate $m_i \sim \mathcal{N}(\mathbf{0}, \mathbf{I})$, $i = 1, 2, 3, 4$, where $m_i$ is i.i.d. 100-dimensional random vector;

Step 2: Generate $m_i \leftarrow (1 - w) \cdot m_i + w \cdot m_1$ for $i = 2, 3, 4$

Step 3: Generate the labels as follows. First, add the four modality vectors and calculate the sum of coordinates. Then, obatin the 1-dimensional label, i.e. $y = (m_1 + m_2 + m_3 + m_4).sum(dim = 1)$

We can see that information from $m_1$ is shared across different modalities, and $w$ controls how much is shared, which one can use to measure the degree of overlaps. We tune it in $\{0.0, 0.2, 0.5, 0.8, 1.0\}$. A high weight $w$ (close to 1) indicates a high degree of overlaps, while $w = 0$ means each modality is totally independent and is non-overlapping.

**Training Setting.** We use the multi-layer perceptron neural network as our model. To be more specific, we first use a linear layer to encode the input to a 10-dimension latent space and then we map the latent feature to the output space. The first layer's input dimension depends on the number of modality. For example, if we use two modalities, the input dimension is 200. We use SGD as our

optimizer, and set a learning rate 0.01, momentum 0.9, batch size 10000, training for 10000 steps. The Mean Square Error (MSE) loss is considered for evaluation in this regression problem.

$\eta$ **vs Modality Correlation** Our aim is to discover the influence of modality correlation on the latent representation quality $\eta$. To this end, Table 4 shows the $\eta$ with the varying number of modalities under different correlation conditions, which is measured by the MSE loss of the pretrain+finetuned modal. The trend that the loss decreases as the number of modalities increases is described in Figure 2(b), which also validates our analysis of Theorem 2. Moreover, Figure 2(b) shows that higher correlation among modalities achieves a lower loss for $\eta$, which means a better latent representation. This emphasizes the role of latent space to exploit the intrinsic correlations among different modalities.

Table 4: Latent representation quality among different correlation situations on synthetic data

| Modalities | MSE Loss (Degree of Overlap ) | | | | |
|---|---|---|---|---|---|
| | 1 | 0.8 | 0.5 | 0.2 | 0.0 |
| $m_1$ | 0 | 12.04±0.39 | 75.89±1.28 | 193.28±1.08 | 301.92±7.85 |
| $m_1, m_2$ | 0 | 8.16±0.17 | 51.25±1.06 | 129.81±4.36 | 207.45±4.68 |
| $m_1, m_2, m_3$ | 0 | 4.18±0.05 | 26.06±0.69 | 65.17±1.52 | 103.23±0.61 |
| $m_1, m_2, m_3, m_4$ | 0 | 0 | 0 | 0 | 0 |

## 6 Discussion

It has been discovered that the use of multi-modal data in practice will degrade the performance of the model in some cases[43, 19, 38, 15]. These works identify the causes of performance drops of multi-modal as interactions between modalities in the training stage and try to improve the performance by proposing new optimization strategies. Therefore, to theoretically understand them, we need to analyze the training process from the *optimization* perspective. Our results, on the other hand, mainly focus on the generalization side, which is separated from optimization and assumes that we get the best performance possible in training. Moreover, our theory is general and does not require additional assumptions on the relationship across every single modality, which may be crucial for theoretically analyzing the observations in a multi-modal performance drop. Understanding why multi-modal fails in practice is a very interesting direction and worth further investigation in future research.

## 7 Conclusion

In this work, we formulate a multi-modal learning framework that has been extensively studied in the empirical literature towards rigorously understanding why multi-modality outperforms single since the former has access to a better latent space representation. The results answer the two questions: when and why multi-modal outperforms uni-modal jointly. To the best of our knowledge, this is the first theoretical treatment to explain the superiority of multi-modal from the generalization standpoint.

The key takeaway message is that the success of multi-modal learning relies essentially on the better quality of latent space representation, which points to an exciting direction that is worth further investigation: to find which encoder is the bottleneck and focus on improving it. More importantly, our work provides new insights for multi-modal theoretical research, and we hope this can encourage more related research.

**Acknowledgments and Disclosure of Funding**

The work of Yu Huang and Longbo Huang is supported in part by the Technology and Innovation Major Project of the Ministry of Science and Technology of China under Grant 2020AAA0108400 and 2020AAA0108403.

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
