# What Makes Multi-modal Learning Better than Single (Provably)

**Yu Huang**[1,*], **Chenzhuang Du**[1,*], **Zihui Xue**[2], **Xuanyao Chen**[3,4],

**Hang Zhao**[1,4], **Longbo Huang**[1,†]

[1] Institute for Interdisciplinary Information Sciences, Tsinghua University
[2] The University of Texas at Austin   [3] Fudan University
[4] Shanghai Qi Zhi Institute

# Appendices

## A   Proof of Main Results

### A.1   Proof of Theorem 1

*Proof.* Let $h'_{\mathcal{M}}$ denote the minimizer of the population risk over $\mathcal{D}$ with the representation $\hat{g}_{\mathcal{M}}$, then we can decompose the difference between $r\left(\hat{h}_{\mathcal{M}} \circ \hat{g}_{\mathcal{M}}\right) - r\left(\hat{h}_{\mathcal{N}} \circ \hat{g}_{\mathcal{N}}\right)$ into two parts:

$$r\left(\hat{h}_{\mathcal{M}} \circ \hat{g}_{\mathcal{M}}\right) - r\left(\hat{h}_{\mathcal{N}} \circ \hat{g}_{\mathcal{N}}\right) \tag{1}$$

$$= \underbrace{r\left(\hat{h}_{\mathcal{M}} \circ \hat{g}_{\mathcal{M}}\right) - r\left(h'_{\mathcal{M}} \circ \hat{g}_{\mathcal{M}}\right)}_{J_1} + \underbrace{r\left(h'_{\mathcal{M}} \circ \hat{g}_{\mathcal{M}}\right) - r\left(\hat{h}_{\mathcal{N}} \circ \hat{g}_{\mathcal{N}}\right)}_{J_2} \tag{2}$$

$J_1$ can further be decomposed into:

$$J_1 = \underbrace{r\left(\hat{h}_{\mathcal{M}} \circ \hat{g}_{\mathcal{M}}\right) - \hat{r}\left(\hat{h}_{\mathcal{M}} \circ \hat{g}_{\mathcal{M}}\right)}_{J_{11}} + \underbrace{\hat{r}\left(\hat{h}_{\mathcal{M}} \circ \hat{g}_{\mathcal{M}}\right) - \hat{r}\left(h'_{\mathcal{M}} \circ \hat{g}_{\mathcal{M}}\right)}_{J_{12}} \tag{3}$$

$$+ \underbrace{\hat{r}\left(h'_{\mathcal{M}} \circ \hat{g}_{\mathcal{M}}\right) - r\left(h'_{\mathcal{M}} \circ \hat{g}_{\mathcal{M}}\right)}_{J_{13}} \tag{4}$$

$$\tag{5}$$

Clearly, $J_{12} \leq 0$ since $\hat{h}_{\mathcal{M}}$ is the minimizer of the empirical risk over $\mathcal{D}$ with the representation $\hat{g}_{\mathcal{M}}$. And $J_{11} + J_{13} \leq 2\sup_{h \in \mathcal{H}, g_{\mathcal{M}} \in \mathcal{G}_{\mathcal{M}}} |r(h \circ g_{\mathcal{M}}) - \hat{r}(h \circ g_{\mathcal{M}})|$.

$$\sup_{h \in \mathcal{H}, g_{\mathcal{M}} \in \mathcal{G}_{\mathcal{M}}} |\hat{r}\left(h \circ g_{\mathcal{M}}\right) - r\left(h \circ g_{\mathcal{M}}\right)|$$

$$= \sup_{h \in \mathcal{H}, g_{\mathcal{M}} \in \mathcal{G}_{\mathcal{M}}} \left| \frac{1}{m} \sum_{i=1}^{m} \ell\left(h \circ g_{\mathcal{M}}(\mathbf{x}_i), y_i\right) - \mathbb{E}_{(\mathbf{x}', y') \sim \mathcal{D}}\left[\ell\left(h \circ g_{\mathcal{M}}(\mathbf{x}'), y'\right)\right] \right|$$

---

*equal contribution

†Correspondence to: longbohuang@tsinghua.edu.cn

35th Conference on Neural Information Processing Systems (NeurIPS 2021).

Since $\ell$ is bounded by a constant $C$, we have $0 \leq \ell(h \circ g_{\mathcal{M}}(\mathbf{x}), y) \leq C$ for any $(\mathbf{x}, y)$. As one pair $(\mathbf{x}_i, y_i)$ changes, the above equation cannot change by at most $\frac{2C}{m}$. Applying McDiarmid's[4] inequality, we obtain that with probability $1 - \delta/2$:

$$\sup_{h \in \mathcal{H}, g_{\mathcal{M}} \in \mathcal{G}_{\mathcal{M}}} |\hat{r}(h \circ g_{\mathcal{M}}) - r(h \circ g_{\mathcal{M}})| \tag{6}$$

$$\leq \mathbb{E}_{(\mathbf{x}_i, y_i) \sim \mathcal{D}} \sup_{h \in \mathcal{H}, g_{\mathcal{M}} \in \mathcal{G}_{\mathcal{M}}} \left| \frac{1}{m} \sum_{i=1}^{m} \ell(h \circ g_{\mathcal{M}}(\mathbf{x}_i), y_i) - \mathbb{E}_{(\mathbf{x}', y') \sim \mathcal{D}} [\ell(h \circ g_{\mathcal{M}}(\mathbf{x}'), y')] \right| \tag{7}$$

$$+ C\sqrt{\frac{2 \ln(2/\delta)}{m}} \tag{8}$$

To proceed the proof, we introduce a popular result of Rademacher complexity in the following lemma[2]:

**Lemma 1.** *Let $U, \{U_i\}_{i=1}^{m}$ be i.i.d. random variables taking values in some space $\mathcal{U}$ and $\mathcal{F} \subseteq [a, b]^{\mathcal{U}}$ is a set of bounded functions. We have*

$$\mathbb{E}\left[ \sup_{f \in \mathcal{F}} \left( \mathbb{E}[f(U)] - \frac{1}{m} \sum_{i=1}^{m} f(U_i) \right) \right] \leq 2\mathfrak{R}_m(\mathcal{F}) \tag{9}$$

*Proof of lemma 1.* Denote $\{U_i'\}_{i=1}^{m}$ be ghost examples of $\{U_i\}_{i=1}^{m}$, i.e. $U_i'$ be independent of each other and have the same distribution as $U_i$. Then we have,

$$\mathbb{E}\left[ \sup_{f \in \mathcal{F}} \left( \mathbb{E}[f(U)] - \frac{1}{m} \sum_{i=1}^{m} f(U_i) \right) \right] \tag{10}$$

$$= \mathbb{E}\left[ \sup_{f \in \mathcal{F}} \left( \frac{1}{m} \sum_{i=1}^{m} (\mathbb{E}[f(U)] - f(U_i)) \right) \right] \tag{11}$$

$$\overset{(a)}{=} \mathbb{E}\left[ \sup_{f \in \mathcal{F}} \left( \frac{1}{m} \sum_{i=1}^{m} \mathbb{E}[f(U_i') - f(U_i) \mid \{U_i\}_{i=1}^{m}] \right) \right] \tag{12}$$

$$\leq \mathbb{E}\left[ \mathbb{E}\left[ \sup_{f \in \mathcal{F}} \left( \frac{1}{m} \sum_{i=1}^{m} (f(U_i') - f(U_i)) \right) \mid \{U_i\}_{i=1}^{m} \right] \right] \tag{13}$$

$$\overset{(b)}{=} \mathbb{E}\left[ \sup_{f \in \mathcal{F}} \left( \frac{1}{m} \sum_{i=1}^{m} (f(U_i') - f(U_i)) \right) \right] \tag{14}$$

$$= \mathbb{E}\left[ \sup_{f \in \mathcal{F}} \left( \frac{1}{m} \sum_{i=1}^{m} \sigma_i (f(U_i') - f(U_i)) \right) \right] \tag{15}$$

$$\leq \mathbb{E}\left[ \sup_{f \in \mathcal{F}} \frac{1}{m} \sum_{i=1}^{m} \sigma_i f(U_i') \right] + \mathbb{E}\left[ \sup_{f \in \mathcal{F}} \frac{1}{m} \sum_{i=1}^{m} \sigma_i f(U_i) \right] \tag{16}$$

$$\overset{(c)}{=} 2\mathfrak{R}_m(\mathcal{F}). \tag{17}$$

where $\sigma_1, \ldots, \sigma_m$ is i.i.d. $\{\pm 1\}$-valued random variables with $\mathbb{P}(\sigma_i = +1) = \mathbb{P}(\sigma_i = -1) = 1/2$. $(a)$ $(b)$ are obtained by the tower property of conditional expectation; $(c)$ follows from the definition of Rademacher complexity of $\mathcal{F}$. $\square$

Consider the function class:

$$\ell_{\mathcal{H} \circ \mathcal{G}_{\mathcal{M}}} := \{(\mathbf{x}, y) \mapsto \ell(h \circ g_{\mathcal{M}}(\mathbf{x}), y) \mid h \in \mathcal{H}, g_{\mathcal{M}} \in \mathcal{G}_{\mathcal{M}}\}$$

let $\mathcal{F} = \ell_{\mathcal{H} \circ \mathcal{G}_{\mathcal{M}}}$ in lemma 1, then we have equation (7) can be upper bound by $2\mathfrak{R}_m(\ell_{\mathcal{H} \circ \mathcal{G}_{\mathcal{M}}})$. To directly work with the hypothesis function class, we need to decompose the Rademacher term which

consists of the loss function classes. We center the function $\ell'(h \circ g_\mathcal{M}(\mathbf{x}), y) = \ell(h \circ g_\mathcal{M}(\mathbf{x}), y) - \ell(\mathbf{0}, y)$. The constant-shift property of Rademacher averages[2] indicates that

$$\mathfrak{R}_m(\ell_{\mathcal{H} \circ \mathcal{G}_\mathcal{M}}) \leq \mathfrak{R}_m(\ell'_{\mathcal{H} \circ \mathcal{G}_\mathcal{M}}) + \frac{C}{\sqrt{m}}$$

Since $\ell'$ is Lipschitz in its first coordinate with constant $L$ and $\ell'(h \circ g_\mathcal{M}(\mathbf{0}), y) = 0$, applying the contraction principle[2], we have:

$$\mathfrak{R}_m(\ell'_{\mathcal{H} \circ \mathcal{G}_\mathcal{M}}) \leq 2L\mathfrak{R}_m(\mathcal{H} \circ G_\mathcal{M})$$

Combining the above discussion, we obtain:

$$J_1 \leq 8L\mathfrak{R}_m(\mathcal{H} \circ G_\mathcal{M}) + \frac{4C}{\sqrt{m}} + 2C\sqrt{\frac{2\ln(2/\delta)}{m}}$$

For $J_2$, by the definition of $h'_\mathcal{M}$:

$$J_2 = \inf_{h_\mathcal{M} \in \mathcal{H}} \left[ r\left(h_\mathcal{M} \circ \hat{g}_\mathcal{M}\right) - r\left(\hat{h}_\mathcal{N} \circ \hat{g}_\mathcal{N}\right) \right] \tag{18}$$

$$\leq \sup_{h_\mathcal{N} \in \mathcal{H}} \inf_{h_\mathcal{M} \in \mathcal{H}} \left[ r\left(h_\mathcal{M} \circ \hat{g}_\mathcal{M}\right) - r\left(h_\mathcal{N} \circ \hat{g}_\mathcal{N}\right) \right] \tag{19}$$

$$= \inf_{h_\mathcal{M} \in \mathcal{H}} \left[ r\left(h_\mathcal{M} \circ \hat{g}_\mathcal{M}\right) - r(h^* \circ g^*) \right] - \inf_{h_\mathcal{N} \in \mathcal{H}} \left[ r\left(h_\mathcal{N} \circ \hat{g}_\mathcal{N}\right) - r(h^* \circ g^*) \right] \tag{20}$$

$$= \eta(\hat{g}_\mathcal{M}) - \eta(\hat{g}_\mathcal{N}) \tag{21}$$

$$= \gamma_\mathcal{S}(\mathcal{M}, \mathcal{N}) \tag{22}$$

Finally,

$$r\left(\hat{h}_\mathcal{M} \circ \hat{g}_\mathcal{M}\right) - r\left(\hat{h}_\mathcal{N} \circ \hat{g}_\mathcal{N}\right) \leq \gamma_\mathcal{S}(\mathcal{M}, \mathcal{N}) + 8L\mathfrak{R}_m(\mathcal{H} \circ \mathcal{G}_\mathcal{M}) + \frac{4C}{\sqrt{m}} + 2C\sqrt{\frac{2\ln(2/\delta)}{m}}$$

with probability $1 - \frac{\delta}{2}$. $\qquad\square$

## A.2    Proof of Theorem 2

*Proof.* Let $\tilde{h}_\mathcal{M}$ denote the minimizer of the population risk over $\mathcal{D}$ with the representation $\hat{g}_\mathcal{M}$, then we have:

$$\eta(\hat{g}_\mathcal{M}) \tag{23}$$

$$= r(\tilde{h}_\mathcal{M} \circ \hat{g}_\mathcal{M}) - r(h^* \circ g^*) \tag{24}$$

$$\leq \underbrace{r(\hat{h}_\mathcal{M} \circ \hat{g}_\mathcal{M}) - \hat{r}(\hat{h}_\mathcal{M} \circ \hat{g}_\mathcal{M})}_{J_1} + \underbrace{\hat{r}(\hat{h}_\mathcal{M} \circ \hat{g}_\mathcal{M}) - \hat{r}(h^* \circ g^*)}_{J_2} + \underbrace{\hat{r}(h^* \circ g^*) - r(h^* \circ g^*)}_{J_3} \tag{25}$$

$J_2$ is the centering empirical risk. Following the similar analysis in Theorem 1, we obtain:

$$J_1 + J_3 \leq \sup_{h \in \mathcal{H}, g_\mathcal{M} \in \mathcal{G}_\mathcal{M}} |r(h \circ g_\mathcal{M}) - \hat{r}(h \circ g_\mathcal{M})| + \sup_{h \in \mathcal{H}, g \in \mathcal{G}} |r(h \circ g) - \hat{r}(h \circ g)| \tag{26}$$

$$\leq 4L\mathfrak{R}_m(\mathcal{H} \circ \mathcal{G}_\mathcal{M}) + 4L\mathfrak{R}_m(\mathcal{H} \circ \mathcal{G}) + \frac{4C}{\sqrt{m}} + 2C\sqrt{\frac{2\ln(2/\delta)}{m}} \tag{27}$$

with probability $1 - \delta$. Combining the above discussion yields the result:

$$\eta(\hat{g}_\mathcal{M}) \leq \tag{28}$$

$$4L\mathfrak{R}_m(\mathcal{H} \circ \mathcal{G}_\mathcal{M}) + 4L\mathfrak{R}_m(\mathcal{H} \circ \mathcal{G}) + \frac{4C}{\sqrt{m}} + 2C\sqrt{\frac{2\ln(2/\delta)}{m}} + \hat{L}(\hat{h}_\mathcal{M} \circ \hat{g}_\mathcal{M}, \mathcal{S}) \tag{29}$$

$$\square$$

### A.3 Proof of Proposition 1

*Proof.* With the $l_2$ loss, we have

$$\mathbb{E}_{\mathbf{x},y\sim h^\star\circ g^\star(\mathbf{x})}\{\ell(h\circ g(\mathbf{x}),y)-\ell(h^\star\circ g^\star(\mathbf{x}),y)\}=\mathbb{E}_{\mathbf{x}}\left[\left|\left|\boldsymbol{\beta}^\top\mathbf{A}^\top\mathbf{x}-\boldsymbol{\beta}^{\star\top}\mathbf{A}^{\star\top}\mathbf{x}\right|\right|^2\right]$$

Define the covariance matrix[9] for two linear projections $\mathbf{A}$, $\mathbf{A}'$ as follows:

$$\Gamma(\mathbf{A},\mathbf{A}')=\mathbb{E}_{\mathbf{x}}\left[\begin{array}{cc}\mathbf{A}^\top\mathbf{x}\left(\mathbf{A}^\top\mathbf{x}\right)^\top & \mathbf{A}^\top\mathbf{x}\left(\mathbf{A}'^\top\mathbf{x}\right)^\top\\ \mathbf{A}'^\top\mathbf{x}\left(\mathbf{A}^\top\mathbf{x}\right)^\top & \mathbf{A}'^\top\mathbf{x}\left(\mathbf{A}'^\top\mathbf{x}\right)^\top\end{array}\right]$$

$$=\left[\begin{array}{cc}\mathbf{A}^\top\Sigma\mathbf{A} & \mathbf{A}^\top\Sigma\mathbf{A}'\\ \mathbf{A}'^\top\Sigma\mathbf{A} & \mathbf{A}'^\top\Sigma\mathbf{A}'\end{array}\right]=\left[\begin{array}{cc}\Gamma_{11}(\mathbf{A},\mathbf{A}^\star) & \Gamma_{12}(\mathbf{A},\mathbf{A}^\star)\\ \Gamma_{21}(\mathbf{A},\mathbf{A}^\star) & \Gamma_{22}(\mathbf{A},\mathbf{A}^\star)\end{array}\right] \tag{30}$$

where $\Sigma$ denotes the covariance matrix of the distribution $\mathbb{P}_\mathbf{x}$. Then the *latent representation quality* of $\mathbf{A}$ becomes:

$$\eta(\mathbf{A})=\inf_{\boldsymbol{\beta}:\|\boldsymbol{\beta}\|\leq C_b}\mathbb{E}_{\mathbf{x}}\left[\left|\left|\boldsymbol{\beta}^\top\mathbf{A}^\top\mathbf{x}-\boldsymbol{\beta}^{\star\top}\mathbf{A}^{\star\top}\mathbf{x}\right|\right|^2\right] \tag{31}$$

$$=\inf_{\boldsymbol{\beta}:\|\boldsymbol{\beta}\|\leq C_b}[\boldsymbol{\beta},-\boldsymbol{\beta}^\star]\,\Gamma(\mathbf{A},\mathbf{A}^\star)\,[\boldsymbol{\beta},-\boldsymbol{\beta}^\star]^\top \tag{32}$$

For sufficiently large $C_b$, the constrained minimizer of (32) is equivalent to the unconstrained minimizer. Following the standard discussion of the quadratic convex optimization [3], if $\Gamma_{11}(\mathbf{A},\mathbf{A}^\star)\succ 0$ and $\det\Gamma_{11}(\mathbf{A},\mathbf{A}^\star)\neq 0$, the solution of the above minimization problem is $\boldsymbol{\beta}=\Gamma_{11}(\mathbf{A},\mathbf{A}^\star)^{-1}\Gamma_{12}(\mathbf{A},\mathbf{A}^\star)\boldsymbol{\beta}^\star$, and

$$\eta(\mathbf{A})=\boldsymbol{\beta}^\star\Gamma_{sch}(\mathbf{A},\mathbf{A}^\star)\boldsymbol{\beta}^{\star\top} \tag{33}$$

where $\Gamma_{sch}(\mathbf{A},\mathbf{A}^\star)$ is the Schur complement of $\Gamma(\mathbf{A},\mathbf{A}^\star)$, defined as:

$$\Gamma_{sch}(\mathbf{A},\mathbf{A}^\star) \tag{34}$$

$$=\Gamma_{22}(\mathbf{A},\mathbf{A}^\star)-\Gamma_{21}(\mathbf{A},\mathbf{A}^\star)\Gamma_{11}(\mathbf{A},\mathbf{A}^\star)^{-1}\Gamma_{12}(\mathbf{A},\mathbf{A}^\star) \tag{35}$$

Under the orthogonal assumption, $\hat{\mathbf{A}}_\mathcal{M}$ is nonsingular. Notice that $\hat{\mathbf{A}}_\mathcal{N}$ cannot be orthonormal in our settings. And $\sum$ is also invertible. Therefore, the Schur complement of $\Gamma(\hat{\mathbf{A}}_\mathcal{M},\mathbf{A}^\star)$ exists,

$$\Gamma_{sch}(\hat{\mathbf{A}}_\mathcal{M},\mathbf{A}^\star)=\mathbf{A}^{\star\top}\Sigma\mathbf{A}^\star-\left(\mathbf{A}^{\star\top}\Sigma\hat{\mathbf{A}}_\mathcal{M}\right)\left(\hat{\mathbf{A}}_\mathcal{M}^\top\Sigma\hat{\mathbf{A}}_\mathcal{M}\right)^{-1}\left(\hat{\mathbf{A}}_\mathcal{M}^\top\Sigma\mathbf{A}^\star\right)=\mathbf{0} \tag{36}$$

Hence, $\eta(\hat{\mathbf{A}}_\mathcal{M})=0$. Given the above discussion, we obtain:

$$\gamma_\mathcal{S}(\mathcal{M},\mathcal{N})=\eta(\hat{\mathbf{A}}_\mathcal{M})-\eta(\hat{\mathbf{A}}_\mathcal{N}) \tag{37}$$

$$=0-\inf_{\boldsymbol{\beta}:\|\boldsymbol{\beta}\|\leq C_b}\mathbb{E}_{\mathbf{x}}\left[\left|\left|\boldsymbol{\beta}^\top\hat{\mathbf{A}}_\mathcal{N}^\top\mathbf{x}-\boldsymbol{\beta}^{\star\top}\mathbf{A}^{\star\top}\mathbf{x}\right|\right|^2\right]\leq 0 \tag{38}$$

$\square$

## B   The Composite Framework in Applications

As we stated in Section 3, our model well captures the essence of lots of existing multi-modal methods [1, 6, 7, 11, 10, 8]. Below, we explicitly discuss how these methods fit well into our general model, by providing the corresponding function class $\mathcal{G}$ under each method.

**Audiovisual fusion for sound recognition [6]:**   The audio and visual models map the respective inputs to segment-level representations, which are then used to obtain single-modal predictions, $\mathbf{h}_a$ and $\mathbf{h}_v$, respectively. The attention fusion function $n_{attn}$, ingests the single-modal predictions, $\mathbf{h}_a$ and $\mathbf{h}_v$, to produce weights for each modality, $\boldsymbol{\alpha}_a$ and $\boldsymbol{\alpha}_v$. The single-modal audio and visual predictions, $\mathbf{h}_a$ and $\mathbf{h}_v$, are mapped to $\tilde{\mathbf{h}}_a$ and $\tilde{\mathbf{h}}_v$ via functions $n_a$ and $n_v$ respectively, and fused using the attention weights, $\boldsymbol{\alpha}_a$ and $\boldsymbol{\alpha}_v$. In summary, $g$ has the form:

$$g=\tilde{\mathbf{h}}_{av}=\boldsymbol{\alpha}_a\odot\tilde{\mathbf{h}}_a+\boldsymbol{\alpha}_v\odot\tilde{\mathbf{h}}_v$$

**Channel-Exchanging-Network [11]:** A feature map will be replaced by that of other modalities at the same position, if its scaling factor is lower than a threshold. $g$ in this problem can be formulated as a multi-dimensional mapping $g := (f_1, \cdots, f_M)$, where subnetwork $f_m(x)$ adopts the multi-modal data $x$ as input and fuses multi-modal information by channel exchanging.

**Other Fusion Methods [7, 10, 8, 1]:** Methods in these works can be formulated into the form we mentioned in the example in Section 3. Specifically, recall the example, $g$ has the form: $\varphi_1 \oplus \varphi_2 \oplus \cdots \oplus \varphi_M$, where $\oplus$ denotes a fusion operation, (e.g., averaging, concatenation, and self-attention), and $\varphi_k$ is a deep network which uses each modality data $x^{(k)}$ as input. Under these notations:

- For the early-fusion BERT method in [8], the temporal features are concatenated before the BERT layer and only a single BERT module is utilized. Here, the $\oplus$ is a concatenation function, and $g$ has the form $(\varphi_1, \varphi_2)$.

- [10, 7]discussed different fusion methods by choosing $\oplus$. (i) Max fusion: the $\oplus$ is the maximum function and $g := max\{\varphi_1, \cdots, \varphi_M\}$; (ii) Sum fusion: $g := \sum \varphi_m$; (iii) averaging; (iv) self-attention and so on.

- The fusion section in the survey [1] provides many works which can be incorporated into our framework.

## C   Discussions on Training Setting

Existing works on multi-modal training demonstrates that naively fusing different modalities results insufficient representation learning of each modality [10, 5]. In our experiments, we train our multi-modal model using two methods: (1), naively end-to-end late-fusion training; (2), firstly train the uni-modal models and train a multi-modal classifier over the uni-modal encoders. As shown in Table 1 and Table 2, naively end-to-end training is unstable, affecting the representation learning of each modality, while fine-tuning a multi-modal classifier over trained uni-modal encoders is more stable and the results are more consistent with our theory. Noting that we use the late-fusion framework here, similar to [10, 5].

Table 1: Latent representation quality vs. The number of the sample size on IEMOCAP. In this table, we show the results from naively end-to-end late-fusion training

| Modalities | Test Acc (Ratio of Sample Size) | | | | |
| --- | --- | --- | --- | --- | --- |
| | $10^{-4}$ | $10^{-3}$ | $10^{-2}$ | $10^{-1}$ | 1 |
| T | 23.66±1.28 | 29.08±3.34 | 45.63±0.29 | 48.30±1.31 | 49.93±0.57 |
| TA | **25.06±1.05** | 34.28±4.54 | **47.28±1.24** | 50.46±0.61 | 51.08±0.66 |
| TV | 24.71±0.87 | **38.37±3.12** | 46.54±1.62 | 49.50±1.04 | 53.03±0.21 |
| TVA | 24.71±0.76 | 32.24±1.17 | 46.39±3.82 | **50.75±1.45** | **53.89±0.47** |

Table 2: Latent representation quality vs. The number of the sample size on IEMOCAP. In this table, we fristly train the uni-modal models and train a multi-modal classifier over the uni-modal encoders to get multi-modal results.

| Modalities | Test Acc (Ratio of Sample Size) | | | | |
| --- | --- | --- | --- | --- | --- |
| | $10^{-4}$ | $10^{-3}$ | $10^{-2}$ | $10^{-1}$ | 1 |
| T | 23.66±1.28 | 29.08±3.34 | 45.63±0.29 | 48.30±1.31 | 49.93±0.57 |
| TA | 22.74±1.86 | 35.14±0.38 | 49.15±0.43 | 50.61±0.28 | 51.78±0.08 |
| TV | 23.64±0.07 | 36.64±1.79 | 46.91±0.68 | 48.96±0.47 | 53.24±0.35 |
| TVA | **25.40±1.06** | **40.87±2.47** | **50.67±0.63** | **52.54±0.60** | **54.55±0.29** |