# OpenReview forum: "What Makes Multi-Modal Learning Better than Single (Provably)"
_NeurIPS.cc/2021/Conference — NeurIPS 2021 Poster_

### Official Review · Reviewer_VZFZ · 2021-07-09

**Rating:** 7
**Confidence:** 3

**Summary:**

This paper focuses on proving multimodal learning performs better than unimodal under the most popular multimodal learning framework. Specifically, they proved that in the linear case, learning with multiple modalities achieves a smaller population risk than only using its subset.

**Limitations And Societal Impact:**

The authors adequately addressed the limitations and potential negative social impact of their work.

**Main Review:**

*Originality*: As far as I know, this is the first theoretical paper to try to prove multimodal learning performs better than unimodal. This is an important direction and lack of analysis before.

*Quality*: Authors conduct sufficient theoretical analysis and proof. I don’t find any technical flaws. Meanwhile, the author conducted experiments to validate the conjectures and theoretical results.

*Clarity*: The paper is easy to follow. I only find one typo, e.g., “When the number of *simple* size” on line 190.

*Significance*: This paper could provide new insight for multimodal theoretical research.

*Comments*:  Although this paper theoretically proves that in the linear case multimodal learning better than single, many potential risks may reduce the performance of the multimodal model in practice. Therefore, I think the paper should discuss why the use of multimodal data in practice will reduce the performance of the model in some cases. For example, [1] gives an analysis from the perspective of different model convergence speed. [2][3] explores multi-modal learning from the perspective of uncertainty. [4] removes the bias in multimodal classifiers to avoid some modalities contribute more to the classification results than others.

[1] What makes training multi-modal classification networks hard? In CVPR 2020

[2] Uncertainty-Aware Audiovisual Activity Recognition Using Deep Bayesian Variational Inference? In ICCV 2019

[3] Trusted multi-view classification. In ICLR 2021

[4] Removing Bias in Multi-modal Classifiers: Regularization by Maximizing Functional Entropies. In NeurIPS 2020


**Time Spent Reviewing:**

10

---

> ### Author Response · Authors · 2021-08-10
> **Response to Reviewer VZFZ**
>
> We thank the reviewer for the valuable feedback and we appreciate the positive comments.
>
>
> #### **Q1**:
>
>  I think the paper should discuss why the use of multimodal data in practice will reduce the performance of the model in some cases.
>
> #### **A1**：
> Thank you for the interesting comment.
>
> [1-4] identify the causes of performance drops of multi-modal as interactions between modalities in the training stage and try to improve the performance by proposing new training/optimization strategies. Therefore, to theoretically understanding them,  we need to analyze the training process from the optimization perspective. Moreover, the relationship across every single modality may be crucial for the analysis. Our theory does not make additional assumptions on the relationship among modalities; and mainly focuses on the generalization side of multi-modal learning, which is separated from optimization. Hence, a thorough theoretical understanding of the performance drops of multi-modal in practical settings may be out of the scope of the current work.
>
> We will include the above discussion in the revision. We also agree with the reviewer that understanding why multi-modal fails in practice is a very interesting direction, and will further investigate this topic in our future research.
>
> #### **Q2**:
>
> Typo
>
> #### **A2**：
>
> Thanks for pointing it out. We will correct it in the revision.

---

### Official Review · Reviewer_3VyC · 2021-07-15

**Rating:** 7
**Confidence:** 2

**Summary:**

This work introduces a generalization bound for multimodal learning, which shows that, for a sufficiently large sample size, the model risk decreases with an increasing number of considered modalities.
The authors also discuss a novel principle to dictate whether new modalities can be beneficial as a function of the dataset size and considered function classes.
An empirical evaluation on both toy and real-world data validates the theory, underlining the benefits of learning from different modalities and laying the foundations for future research.

**Limitations And Societal Impact:**

The paper has a strong theoretical focus and does not mention any limitation or societal impact of the proposed analysis.



**Main Review:**

The paper introduces a formal definition of the multimodal learning problem, clearly stating the initial assumptions and introducing the concept of "latent representation quality" to show under which condition learning can benefit from multiple modalities.
The paper is clear and well-written, and all the main results are clearly stated and proved.
To the best of the reviewer's knowledge, the analysis reported in this paper is novel and relevant to the field of multimodal learning.

Main comments:

1) Can the authors further comment on the role of the intrinsic properties mentioned in the principle in lines 189-190 with respect to the modality selection procedure? What role does the change in the size of the possible representation space $\mathcal{G}_{\mathcal{N}}$ play?

2) The data generation process described in lines 277-281 is unclear. How is the 1-dimensional label defined? What procedure used to correlate the modalities? Further details would improve clarity and reproducibility.

3) Reducing the number of available modalities can be considered as a restriction of the size of the hypothesis class. How does the proposed analysis relate to classic PAC bounds in learning theory [1]?

Other Comments and general questions:

4) According to the definition, for a sufficiently large $\mathcal{H}$ the representation quality seems to be closely related to the amount of predictive information loss: as long as the representation mapping $g$ does not discard predictive information, we can find a task mapping $h$ so that $r(h \circ g) = r(h^* \circ g^*)$. In this sense, the difference of representation quality for $\mathcal{N}\subset\mathcal{M}$ could be interpreted as the amount of task-relevant information loss caused by removing informative modalities: $I({\bf y};p_{\mathcal{M}}({\bf x})) \ge I({\bf y};p_{\mathcal{N}}({\bf x}))$ for $\mathcal{N}\subseteq\mathcal{M}$. On the other hand, the addition of new modalities increases the rademacher complexity and the total amount of information $I({\bf x};p_{\mathcal{M}}({\bf x})) \ge I({\bf x};p_{\mathcal{N}}({\bf x}))$.
Can the authors underline any connection with the Information bottleneck principle? Can the addition of redundant modalities $I({\bf y};p_{\mathcal{M}}({\bf x})) = I({\bf y};p_{\mathcal{N}}({\bf x}))$ ever be favorable according to the proposed view-selection principle.

5) Line 159-160: The expression for the Rademacher complexity of $\mathcal{F}$ should contain $f(Z_i)$ instead of $g(Z_i)$.

References:
[1] Shalev-Shwartz, Shai, and Shai Ben-David. Understanding machine learning: From theory to algorithms. Cambridge university press, 2014.


**Time Spent Reviewing:**

4.5 h

---

> ### Author Response · Authors · 2021-08-10
> **Response to 3VyC**
>
> We thank the reviewer for the insightful and detailed review. We appreciate the positive comments.
>
> (In the following comments, sometimes we use plain fonts instead of calligraphic fonts, e.g. $M$ replaces $\mathcal{M}$, due to the limited capacity of OpenReview's TeX support)
>
> #### **Q1**.
>
> Can the authors further comment on the role of the intrinsic properties mentioned in the principle in lines 189-190 with respect to the modality selection procedure?  What role does the change in the size of the possible representation space $G_N$ play?
>
> #### **A1**
> Using more modalities will expand the representation space，and searching the minimum in a larger space will result in a lower empirical risk. Therefore, the change in the size of the possible representation space $G_{N}$ will affect the optimization of empirical risk.
>
> Regarding the role of the intrinsic properties, $\\mathcal{C}(\cdot)$ captures the complexity of the function class, e.g., VC dimension, absolute dimension for the finite hypothesis class. Generally,   $G_N \\subset G_M$ implies $\\sqrt{C\\left(H\\circ G_{M}\\right)}-\\sqrt{C\\left(H\\circ G_{N}\\right)}\geq 0$,  which is a constant only determined by the chosen modality sets  $\mathcal{M}$ and $\mathcal{N}$.  When  $m\rightarrow \infty$,  i.e., the training data is abundant,  the RHS of the principle $\\sqrt{\frac{C\left(H \circ G_{M}\\right)}{m}}-\sqrt{\frac{C\left(H \circ G_{N}\\right)}{m}}$  will go to zero, and the inequality in the principle holds. Combining this with Theorem 2, we conclude that the latent representation quality of $\hat{g}_{M}$   may be less than $\hat{g}_N$, and tend to use more modalities. On the other hand, when there are not enough samples, the impact of complexity cannot be ignored, and there may be situations where fewer modalities are used for better performance.
>
> #### **Q2**
>
> The data generation process described in lines 277-281 is unclear. How is the 1-dimensional label defined? What procedure used to correlate the modalities? Further details would improve clarity and reproducibility.
>
> #### **A2**
> The code included in *simulate.py* in the supplementary material provides the detailed generating process. For the synthetic data,  we have four modalities, and the generation process is summarized as follows:
>
> Step 1: Generate $m_i\sim \mathcal{N}(\mathbf{0}, \mathbf{I})$​​​​​​​,​​ $i=1,2,3,4$​​​,  where $m_i$​​​ is i.i.d. 100-dimensional random vector;
>
> Step 2: Generate $m_i\leftarrow (1-w)\cdot m_i+w\cdot m_1$​​ for $i=2,3,4$
>
> We can see that information from $m_1$ is shared across different modalities,  and $w$ controls how much is shared, which one can use to measure the degree of overlaps.  A high weight $w$ (close to $1$) indicates a high degree of overlaps, while $w=0$​ means each modality is totally independent  and  is non-overlapping.
>
> We generate the labels as follows. First, add the four modality vectors and calculate the sum of coordinates. Then, obatin the 1-dimensional label, i.e. $y=(m_1+m_2+m_3+m_4).sum(dim=1)$​​​​​.
>
>
> Thanks for noticing this, we will make it clearer in the revision.
>
> #### **Q3**
>
>  Reducing the number of available modalities can be considered as a restriction of the size of the hypothesis class. How does the proposed analysis relate to classic PAC bounds in learning theory ?
>
> #### **A3**
> The proposed analysis has a certain connection with the existing PAC bounds; since they are both based on the uniform convergence framework, a widely used tool in learning theory.  The  classic PAC method requires the hypothesis class to be any type of binary classifier (finite) and uses  complexity measures that depend on the total number of parameters of the network [1]. However, when models in consideration have large parameters, e.g.,  overparametrized neural networks [2], such complexity measures lead to  trivial bounds. Our analysis is more general and applicable,  without strict assumptions on the hypothesis class and complexity measures.
>
> #### **Q4**.
>
>  Can the authors underline any connection with the Information bottleneck principle? Can the addition of redundant modalities $I(y;p_{\mathcal{M}}(x))=I(y;p_{\mathcal{N}}(x))$​​ ever be favorable according to the proposed view-selection principle.
>
> #### **A4**
>
> The idea of using mutual information to measure the latent representation quality when $\mathcal{H}$ is sufficiently large is interesting. Note that in [3], the IB principle is formulated as the maximization of $I(y;z) − βI(x;z )$, which  is to learn an encoding $z$ that is maximally expressive about $y$ while being maximally compressive about $x$.  However, in our work,  $I(y;p_{\mathcal{M}}(x))>I(y;p_{\mathcal{N}}(x))$ and $I(x;p_{\mathcal{M}}(x))>I(x;p_{\mathcal{N}}(x))$  for $\mathcal{N}\subset \mathcal{M}$, i.e.,   learning with more modalities maximizes mutual information with $x$ and $y$ simultaneously, which violates the definition of good representation in IB. Hence we cannot directly apply the IB principle here for understanding the benefit of multi-modal.
>
> $I(y;p_{\mathcal{M}}(x))=I(y;p_{\mathcal{N}}(x))$  indicates that additional modalities cannot increase the task-relevant information, and intuitively training with modality set $\mathcal{N}$ （fewer modalities）is preferable. Hence, adding more modalities cannot further optimize the empirical risk, i.e., $\hat{L}(\hat{h}_N \circ \hat{g}_N, \mathcal{S})-\hat{L}(\hat{h}_M \circ \hat{g}_M, \mathcal{S})$. In this case, the LHS of the view-selection principle will also be very close to 0. Combining this with the fact that both sides of the inequality of our principle are non-negative (through the discussion in line 186-188), this implies that the inequality is violated with high probability and we need to use fewer modalities for learning. Therefore, the modality set selected by our principle is consistent with that directly derived from the view of information theory in this case.
>
> #### **Q5**
>
>  Line 159-160: The expression for the Rademacher complexity of $F$ should contain $f(Z_i)$ instead of $g(Z_i)$.
>
> #### **A5**
> We thank the reviewer for pointing out this typo. We will correct it in the revision.
>
> Reference：
>
> [1] Shai Shalev-Shwartz and Shai Ben-David.Understanding machine learning: From theory toalgorithms. Cambridge university press, 2014.
>
> [2] Behnam  Neyshabur,  Zhiyuan  Li,  Srinadh  Bhojanapalli,  Yann  LeCun,  and  Nathan  Srebro.Towards understanding the role of over-parametrization in generalization of neural networks. 2018
>
> [3] Alexander A Alemi,  Ian Fischer,  Joshua V Dillon,  and Kevin Murphy.  Deep variational information bottleneck. 2016.

---

> > ### Comment · Reviewer_3VyC · 2021-08-23
> > **Thank you for your replies**
> >
> > I thank the authors for their insightful replies and clarifications.
> > I hope the authors will consider including some additional intuition regarding the view-selection criteria and the role of the intrinsic property in the main text.
> > A summary of the data generating procedure described by the authors can also help the reader to better understand the experimental section.

---

> > > ### Author Response · Authors · 2021-08-23
> > > **Reply to Reviewer 3VyC**
> > >
> > > Thank you for the suggestion! We will certainly elaborate on the data generating procedure and include additional intuition inspired by
> > > the reviewer’s insightful reviews in our final version.

---

### Official Review · Reviewer_eSGN · 2021-07-16

**Rating:** 4
**Confidence:** 4

**Summary:**

This paper aims to provide a theoretical explanation on whether multimodal learning can perform better than unimodal learning. They study a popular multimodal learning framework which firstly encodes features from different modalities into a common latent space before mapping the latent representations into a label. They prove that under certain conditions multimodal learning enables better estimation of the latent space which then enables better task performance. They also show some experiment results to back up their theory.

**Ethical Concerns:**

There are no ethical concerns as far as I can tell.

**Limitations And Societal Impact:**

I think the authors can say a bit more about the societal impact of multimodal models, including their applications in robotics, healthcare, education etc.

**Main Review:**

Strengths:
1. The problem tackled is an important one and an important shortcoming in current work.
2. There are some empirical results supporting their results.

Weaknesses:
1. There are some severe issues with the setup, results, and writing which makes the paper hard to follow and understand:

a. In line 38 you say you analyze 'a widely used composite multimodal framework [35]' but [35] was only proposed in 2020 so I'm not sure how it can be deemed as a 'widely used' framework.

b. Also [35] uses a SVM on each modality's (total M modalities) features to get them into a common latent space from which a single classifier is used from that space to the label, and each modality is used for predicting the label to get M labels, from which you take an ensemble to predict the final label. But in your setup you don't do this, and the example you give in line 124-128 talks about self-attention to combine the features. I'm not sure why there is inconsistency here.

c. In line 115 I'm not sure why noisy modalities are being introduced. If you want to analyze multimodal learning why don't you start by analyzing the case where all modalities are present. I'm also not sure what p'_M means as an extension of p_M - are you adding noisy modalities on top of noisy modalities? I'm not sure what is going on here.

2. There are many multimodal datasets that can be used for research, but this paper only uses 1 dataset and a synthetic dataset. You can refer to https://arxiv.org/abs/2107.07502 for a set of comprehensive benchmarks to test on.

3. There are severe clarity issues with the writing such that I have a hard time understanding the experimental results.

a. The experiments are on concatenating features which seems to be different from your analysis and also different from [35].

b. line 245: 'freeze the encoder gˆM obtained 246 through pretraining and then finetune a better classifier h.' I'm not sure what this means.

c. Many of the figure captions are not informative (e.g. figure 2) which makes it hard to understand.

d. Many details are missing in the synthetic data, including what exactly 'overlap' means and how the labels are generated.

4. It would be good to refer to 'What Makes Training Multi-Modal Classification Networks Hard? https://arxiv.org/abs/1905.12681' as an example of where simple multimodal learning (i.e. concatenating features) fails. I believe the theory proposed here is unable to explain this phenomenon. I'm also not sure if the assumptions made in this paper are too strict or too loose to reconcile these empirical findings.

Overall this paper would benefit from a significant revision. I would suggest carefully reading the multimodal literature (see https://arxiv.org/abs/2107.07502 and https://arxiv.org/abs/1705.09406) for comprehensive models, benchmarks, and surveys. It would also be good to see under what settings do multimodal models fail (like in https://arxiv.org/abs/1905.12681 or https://aclanthology.org/2020.emnlp-main.62.pdf) because just by looking at methods that work will make it hard to come up with an explanation that is consistent with both sides of the story. The writing and clarity of this paper can also be significantly improved.

**Time Spent Reviewing:**

3 hours

---

> ### Author Response · Authors · 2021-08-10
> **Response to Reviewer eSGN**
>
> We thank the reviewer for the time and thoughtful comments on our work.
>
> (In the following comments, sometimes we use plain fonts instead of calligraphic fonts, e.g.  replaces , due to the limited capacity of OpenReview's TeX support)
>
> We would like to emphasize that the main focus of the paper is on developing a novel mathematical theory towards rigorously understanding why multi-modality outperforms single. In particular, we propose a novel composite framework that firstly encodes the complex data from heterogeneous sources into a common latent space, and then introduce an innovative metric latent representation quality to show under which condition learning with multiple modalities can achieve a smaller population risk than only using its subset of modalities.
>
> #### **Q1**
>
> There are some severe issues with the setup, results, and writing which makes the paper hard to follow and understand:
>
> #### **A1**
>
> We firstly respond to your concerns **Q1.a** and **Q1.b** together:
>
> **A1.a, A1.b**
>
> We cited [35] because it formally states the existence of latent space and the composite structure, which motivates our formulation and study. Therefore, we cite [35] after the claim 'composite multimodal framework ' to provide the evidence that it has a related definition in the literature.
>
> Note that the composite multimodal framework is often observed in applications. In fact, in recent years,  a large number of papers, e.g., [1-6], appear to have utilized this framework in one way or another, even though the contributors did not clearly summarize the relationship between their methods and this common underlying structure. However, despite the popularity of the framework, existing works lack a very formal definition in theory.
>
> Thank you for pointing out the inaccuracies here. To avoid confusion, we will clarify these points in the revision.
>
> **A1.c**.
>
> There might be a misunderstanding about the use of '$\perp$' here. Also, noisy modality is not necessary for our analysis.
> In line 115, we explain that  $\mathbf{x}_{k}^{\prime}=\perp$ means that the k-th modality is not used. The original $p_M$ is a mapping to indicate what modalities are used. Since $\mathcal{X}\subset\mathcal{X}^{\prime}$, by definition, we can naturally extend the input of $p_M$ from $\mathcal{X}$  to $\mathcal{X}^{\prime}$ and obtain $p^{\prime}_M$, which is actually the  identity transformation on $\mathcal{X}^{\prime}$, i.e., $p^{\prime}_M(\mathbf{x}^{\prime})= \mathbf{x}^{\prime}$ for any $\mathbf{x}^{\prime}\in \mathcal{X}^{\prime}$. Our purpose of defining the notations in this way is for convenience of the subsequent assumption description from the mathematical perspective, and is not related to the structure of the multi-modal.
>
> #### **Q2**
>
> There are many multimodal datasets that can be used for research, but this paper only uses 1 dataset and a synthetic dataset.
>
>
> #### **A2**
> We would like to emphasize that the main focus of the paper is on developing a novel mathematical theory towards rigorously understanding why multi-modality outperforms single. Empirical evaluations were presented to help us validate important properties of the theory. On the other hand, we agree with the reviewer that the experimental results could be strengthened, and will certainly look into the datasets suggested by the reviewer.
>
> #### **Q3**
>
> There are severe clarity issues with the writing such that I have a hard time understanding the experimental results.
>
> #### **A3**
>
> **A3.a**. Recall the example at the end of Section 3, where $g_{\mathcal{M}}$ has the form: $\varphi_{1} \oplus \varphi_{2} \oplus \cdots \oplus \varphi_{M}$, where $\oplus$ denotes a fusion operation, and $\varphi_{k}$ is a deep network.​Let $\oplus$​here be the concatenating function, which is a common fusion technique. We regard the fully concatenating vector space as the latent space,  and the single latent feature vector of Unimodal can be treated as the special concatenating vector with some coordinates missing (or equal to zero). Then, our framework  immediately applies to this setting.  Moreover, as we explained in bullet **A1**,  our framework and the model in [35] are not completely consistent.
>
> **A3.b.** In our framework, the latent representation quality is defined as the best achievable population risk with the fixed latent representation $g$​ in line 158. To evaluate this property  for a latent mapping $\hat{g}_M$​(encoder) in practice,  we first obtain $\hat{g}_M$​ through pre-traing. In the subsequent training process, we freeze  weights of layers corresponding to $\hat{g}_M$,​  and only train the weights of layers corresponding to classifiers $h$​  in the network to get an approximation of $h^{\prime}$​, where $h^{\prime}$​ minimizes the emprical risk with the fixed $\hat{g}_M$​, i.e. $r(h^{\prime}\circ\hat{g}_M)=\\inf r(h\circ\hat{g}_M)$​​​​ for $h\in\mathcal{H}$.
>
> **A3.c.** We simplify the captions due to limited space constraints. But we did provide  detailed discussions about the figures in Section 5. For Fig 2(a), in lines 258-261, we state that it plots the corresponding curve of Table 3, which  presents  the latent representation quality $\eta$ obtained by using different numbers of sample. We also point out that  the number of used sample is normalized by the total number of training samples. Also, for Fig 2(b), in line 291-291, we illustrated that it shows the trends of Table 4, which measure the  latent representation quality $\eta$ by the MSE-loss with a varying number of modalities under  different correlation conditions.
>
> **A3.d.**   The code  included in *simulate.py* in the supplementary material provides the detailed generated process. For the synthetic data,  we have four modalities, and the generation is summarized as follows:
>
> Step 1: $m_i\sim \mathcal{N}(\mathbf{0}, \mathbf{I})$​​​​​​​,​​ $i=1,2,3,4$​​​,  where $m_i$​​​ is i.i.d. 100-dimensional random vector;
>
> Step 2: $m_i\leftarrow (1-w)\cdot m_i+w\cdot m_1$​​ for $i=2,3,4$
>
> We can see that information from $m_1$ is shared across  different modalities,  and $w$ controls how much is shared, which one can use to measure the degree of overlaps.  A high weight $w$ (close to $1$) indicates a high degree of overlaps, while $w=0$​ means each modality is totally independent  and  is non-overlapping.
>
> We generate the labels as follows. First, add the four modality vectors and  calculate the sum of coordinates. Then, obatin the 1-dimensional label, i.e. $y=(m_1+m_2+m_3+m_4).sum(dim=1)$
>
> #### **Q4**
>
>  I believe the theory proposed here is unable to explain this phenomenon [5]. I'm also not sure if the assumptions made in this paper are too strict or too loose to reconcile these empirical findings.
>
> #### **A4**
> We would like to emphasize that the main focus of the paper is on developing a novel mathematical theory towards rigorously understanding why multi-modality outperforms single. To achieve this goal, we formulated a model movitated by the multimodal fusion problem, which dates back 25 years ago [1] and has been extensively studied in the literature, and begin with a few mild assumptions. Specifically, Assumptions 1-2 are classical conditions in learning theory. Also, Assumption 3 is made for simplicity in mathematical analysis and relates little to the multi-modal itself.
>
> Reference [5] (suggested by the reviewer) focuses on the interesting hardness problem of training in multi-modal learning. It empirically demonstrates an insightful result: the difference in convergence rates of different modalities is the main source of the problem. As a result, joint training will cause overfitting to the modality with fast rate. To theoretically understanding this issue, one needs to consider the training process from the optimization perspective.
>
> Our results, on the other hand, mainly focus on the generalization side, which is seperated from optimization and assumes that we get the best performance possible in training. This is an equally important aspect of multi-modality learning and can be viewed as complementary to that studied in [5]. Moreover, our theory is general, and does not require additional assumptions on the relationship across each single modality, which may be crucial for theoretically analyzing the observations in [5].
>
> We agree with the reviwer that a thorough theoretical understanding for both training and generalization is very important and worth further investigation, and plan to study the topic in our future research.
>
>
> Reference:
>
> [1] Tadas Baltrušaitis, Chaitanya Ahuja, and Louis-Philippe Morency. Multimodal machine learning: A survey and taxonomy.IEEE transactions on pattern analysis and machine intelligence,41(2):423–443, 2018
>
> [2] Haytham M Fayek and Anurag Kumar. Large scale audiovisual learning of sounds with weaklylabeled data. 2020
>
> [3] Christoph Feichtenhofer, Axel Pinz, and Andrew Zisserman. Convolutional two-stream networkfusion for video action recognition. InProceedings of the IEEE conference on computer visionand pattern recognition, pages 1933–1941, 2016
>
> [4] Yikai Wang, Wenbing Huang, Fuchun Sun, Tingyang Xu, Yu Rong, and Junzhou Huang. Deep multimodal fusion by channel exchanging. Advances in Neural Information Processing Systems,33, 2020.
>
> [5] Weiyao Wang, Du Tran, and Matt Feiszli. What makes training multi-modal classification networks hard? In 2020 IEEE/CVF Conference on Computer Vision and Pattern Recognition (CVPR), pages 12692–12702. IEEE, 2020.
>
>
> [6] M Esat Kalfaoglu, Sinan Kalkan, and A Aydin Alatan. Late temporal modeling in 3d cnn architectures with bert for action recognition. In European Conference on Computer Vision, 346 pages 731–747. Springer, 2020.

---

> > ### Comment · Reviewer_eSGN · 2021-08-28
> > **thank you for your response - issues still remain**
> >
> > Thank you authors for your detailed response to my comments. Let me start by noting that I really like this direction of explaining multimodal learning, a largely empirical field that would benefit from rigorous theoretical analysis. However, any paper published in this direction should be rigorous and aligned with the empirical insights. While this paper presents some ideas in this direction, I believe it still has severe issues that would hugely benefit from another round of revision.
> >
> > What exactly are the authors trying to analyze? Despite the efforts in the paper and in the rebuttal, I still do not understand. A 'latent space' and 'composite structure' and 'composite multimodal framework' are overloaded terms and not well-defined. The authors also cite [35] as motivation, but this is not an established or widely-used work. As far as I know all the references [1-6] are different so I'm not sure what the authors are getting at when they say they all utilize this framework. [1] is even a survey paper that cites more than 200 relevant papers in the field. [2,3,4] are all fusion-based methods but all do so differently, and are also different from the 'concatenation' method which you mention in A3.
> >
> > The reason I am so pedantic on the exact definition/structure of the model is because, in my opinion, one cannot perform useful theoretical and empirical analysis without knowing what the exact algorithm is. It is well-known (and clear) that if the multiple modalities give information useful for the label the presence of more modalities will improve performance. But the central problem of multimodal learning lies in 1) understanding and obtaining these useful data sources, 2) defining the model to learn from heterogeneous data (the bulk of the review in survey paper [1]), and 3) evaluating and analyzing multimodal representations. In particular, for point 2, it is also well-known that despite each modality being useful, certain fusion and representation methods works better than others (again see survey paper [1] and empirical reviews like https://arxiv.org/abs/2107.07502). There can be differences in performance even when minor changes made to the fusion method (latent space learning as you call it). Therefore I am really puzzled about this theoretical analysis - while it does formalize some existing well-known empirical observations, it does so without bringing my any insight on how to improve/analyze/debug multimodal learning problems.
> >
> > That being said, I like this direction, and would love to see improvements in the following areas:
> > 1. What are the takeaways? A main part of the analysis lies in using a definition of latent representation quality to analyze the multimodal latent space, and that more modalities used to estimate this latent representation improves its quality. However in the real-world this quality depends extremely heavily on the types of modalities used and the method of learning this latent space. A useful analysis should take this into account, so there can be insights such as certain methods working better for certain datasets, when to use certain modalities, etc. Right now the analysis, in my opinion, remains simplistic and a bulk of it is derived from standard analysis of sample complexities.
> >
> > 2. What is special about multimodal learning? The core challenge lies in the heterogeneity between your modalities (images/text, spatial vs temporal data, set-based vs tabular data). This analysis does not mention any source/analysis of heterogeneity at all, and the analysis could very well be applied to an image classification problem where an image is broken into 3x3=9 image patches and these represent the 9 modalities used as input to a model. In the real-world this core heterogeneity challenge is the main motivation for multimodal datasets and models, but this paper simplifies the problem too much in my opinion.

---

> > > ### Author Response · Authors · 2021-08-30
> > > **Response to Reviewer eSGN**
> > >
> > > We thank the reviewer for the additional comments. We note from the comments that there are still some potential misunderstandings, and will explain them in detail below.
> > >
> > > We first would like to emphasize that our main contribution in this work is a *precise* mathematical formalization of existing multimodal problems. Specifically, we provide generalization bounds for learning with different modalities, to understand when and why multimodal outperforms unimodal jointly.
> > >
> > > In learning theory area, the classic view divides the research problems into three categories[1]:
> > >
> > > **Optimization**: focus on how one can optimize the loss functions to achieve small training error
> > >
> > > **Generalization**: focus on controlling the gap between training and testing error
> > >
> > > **Approximation**:  focus on whether there exists a deep network that achieves low error over the distribution.
> > >
> > > As we pointed out in the rebuttal, our analysis mainly focuses on the generalization side and concerns the learning performance under the best model training. This is important as it provides a fundamental understanding of how different modalities could impact the ultimate learning performance.
> > >
> > > However,  what the reviewer suggested in the comments regarding the training process, e.g., ‘certain fusion and representation methods work better than others', ‘quality depends extremely heavily on the types of modalities used’,  ‘the method of learning this latent space’,  ‘certain methods working better for certain datasets’ and ‘ when to use certain modalities’,  all fall into the aspect of **optimization**,  which is different from and separated from generalization.
> > >
> > > #### **Q**. What exactly are the authors trying to analyze?
> > > #### **A**.
> > > In our work, we formulate and analyze a multimodal framework, which first encodes the complex data from heterogeneous sources into a common latent space, and then implements the specific task on this latent space. Our framework is motivated by the fusion problem, which is one of the most researched aspects of multimodal machine learning with work dating to 25 years ago. In technical terms, it first encodes and fuses information from multiple modalities, which can be regarded as the latent representation mapping $g$, and then passes the fused result to a task-specific function, to predict an outcome measure of a class through classification, or a continuous value through regression, which corresponds to our task function $h$.  We frame the encoding process as a mapping that depends on a general function class $\mathcal{G}$, abstracting those fusion-based encoding methods.
> > >
> > > We emphasize that our model well captures the essence of the methods listed in references [1-6]. Below, we explicitly discuss how these methods fit well into our general model, by providing the corresponding function class $\mathcal{G}$ under each method.
> > >
> > > 1. Audiovisual fusion for sound recognition in [2]: The audio and visual models map the respective input to segment-level representations, which are then used to obtain single-modal predictions, $h_a$ and $h_v$, respectively. The attention fusion function $n_{attn}$, ingests the single- modal predictions, $h_a$ and $h_v$, to produce weights for each modality, $\alpha_{a}$ and $\alpha_{v}$. The single-modal audio and visual predictions, $h_a$ and $h_v$, are mapped to $f_a$and  $f_v$ via functions $n_{a}$ and $n_{v}$ respectively, and fused using the attention weights, $\alpha_{a}$ and $\alpha_{v}$. In summary, $g$ has the form: $g =h_{av}=\alpha_{a} \odot f_{a}+\left(1-\
> > > \alpha_{a}\right) \odot f_{v}$
> > >
> > > 2. Channel-Exchanging-Network in [4]:  A feature map will be replaced by that of other modalities at the same position, if its scaling factor is lower than a threshold. $g$ in this problem can be formulated as a multi-dimensional mapping $g:=(f_1,\cdots,f_M)$, where subnetwork $f_m(x)$ adopts the multi-modal data $x$ as input and fuses multimodal information by channel exchanging.
> > >
> > > 3. Methods in [3][5][6]: They can be formulated into the form we mentioned in our reply in A3.a.  Specifically, recall our reply in A3.a, $g$ has the form: $\varphi_{1} \oplus \varphi_{2} \oplus \cdots \oplus \varphi_{M}$, where $\oplus$ denotes a fusion operation, (e.g., averaging, concatenation, and self-attention), and $\varphi_{k}$ is a deep network which uses each modality data $x^{(k)}$ as input. Under these notations,
> > >
> > >     *a*. For the early-fusion BERT method in [6], the temporal features are concatenated before the BERT layer and only a single BERT module is utilized. Here, the $\oplus$ is a concatenation function, and $g$ has the form $(\varphi_{1} ,\varphi_{2} )$.
> > >
> > >    *b*. [3], [5] discussed different fusion methods by choosing $\oplus$. (i) Max fusion: the $\oplus$ is the maximum function and $g:=\max (\varphi_{1},\cdots, \varphi_{M})$; (ii) Sum fusion:  $g:=\sum \varphi_{m}$; (iii) averaging; (iv) self-attention and so on.
> > >
> > > 4. The fusion section in the survey [1] provides many works which can be incorporated into our framework.
> > >
> > > #### **Q**. One cannot perform useful theoretical and empirical analysis without knowing what the exact algorithm is.
> > > #### **A**.
> > > 1. The reviewer raised some important points about the central problem of multimodal learning. However, the statement that ‘It is well-known (and clear) that if the multiple modalities give information useful for the label the presence of more modalities will improve performance’ does not provide a rigorous and precise understanding of how modalities impact performance. As far as we know, even this seemingly ‘clear’ and fundamental question has no rigorous theoretical guarantee, which is exactly what we analyzed and contributed in our paper.
> > > 2. We emphasize again that we focus on the **generalization** side in our work under a general multi-modality learning model. In the classic theoretical literature focusing on generalization, e.g., [2][3], the analysis is often carried out assuming the best performance possible in the training process (which can be viewed as orthogonal to optimizing the training algorithms).  We agree with the reviewer that the types of modalities used and the method of learning the latent space are very important and worth further investigation. But to theoretically understand these issues, one needs to consider the training process from the optimization perspective, which is outside the scope of our paper.
> > >
> > > #### **Q**. What are the takeaways?
> > > #### **A**.
> > > The key takeaway message we show is that the success of multimodal learning relies essentially on the better quality of latent space representation. Specifically, we prove that the performance of multimodal learning in terms of population risk can be bounded by the latent representation quality, and provide an upper bound for the latent representation quality of training over a subset of modalities.
> > >
> > > As we mentioned in Section 6, our results provide insights on interesting directions that are worth further investigation. For instance,  to find out which modality encoder is the bottleneck and focus on improving it, which is consistent with the reviewer’s suggestions on when to use certain modalities.  We believe that such more specific problems as the reviewer suggested need more assumptions on the relationship between modalities and mostly rely on investigating the optimization process. Our generalization-side results, on the other hand, shed new light on rigorously understanding the fundamental roles of different modalities.
> > >
> > > #### **Q**. What is special about multimodal learning? And heterogeneity between modalities
> > > #### **A**.
> > > We emphasize that we have indeed considered the source of heterogeneity in our paper. In particular,
> > > 1. As we defined in Line 107, a multi-modal representation $\mathbf{x}:=\left(x^{(1)}, \cdots, x^{(K)}\right)$ consists of $K$ modalities. Here, we explicitly allow the dimension of the domain set of each modality $\mathcal{X}^{(k)}$ to be different, which well models the heterogeneity of each modality.
> > > 2. We clearly know that the relationships across different modalities are usually of varying levels due to the heterogeneous sources. Therefore, as a starting point, we make no assumptions on the relationship across every single modality in our analysis, which makes it general to allow different correlations.
> > >
> > > [1] Poggio, Tomaso and Banburski, Andrzej and Liao, Qianli. Theoretical issues in deep networks, Proceedings of the National Academy of Sciences 2020
> > >
> > > [2] Nilesh Tripuraneni, Michael I. Jordan, Chi Jin. On the Theory of Transfer Learning: The Importance of Task Diversity, Neurips 2020
> > >
> > > [3]Sanjeev Arora, Hrishikesh Khandeparkar, Mikhail Khodak, Orestis Plevrakis, Nikunj Saunshi. A Theoretical Analysis of Contrastive Unsupervised Representation Learning, ICML 2019

---

### Official Review · Reviewer_zJa7 · 2021-07-19

**Rating:** 6
**Confidence:** 4

**Summary:**

The paper tries to prove that learning with multiple modalities can achieve a smaller population risk than only using its subset of modalities.

**Limitations And Societal Impact:**

Yes

**Main Review:**

I am glad to see the work which aims to answer the question: Can multimodal learning provably performs better than unimodal? Under a specific setting, the authors provide theoretical results.

Although under a simple setting, the theoretical results are good. There is another work [1] that has provided some analysis about why multiple modalities are better. I think it is interesting to discuss the relation between the proposed one and [1].

Some minors:
“which points to an interesting direction that are worth further…” should be “which points to an interesting direction that is worth further…”

The authors claim “we provide the first theoretical analysis to shed light on what makes multimodal outperform unimodal…”, “this is the first theoretical treatment to capture important qualitative phenomena observed in real multimodal applications”. Since the CPM-Nets has also provided related theoretical results (from other perspective), it not proper to claim the work as the first one.

[1] CPM-Nets: Cross Partial Multi-View Networks, NeurIPS 2019

The authors claim that “two common principles, consensus and complement, which are crucial in the theoretical analysis of multi-view learning, are not applicable in multimodal learning”. Since in machine learning, multimodal and Multiview learning are quite similar, so the claim is not reasonable. Furthermore, the description for the difference between multimodal learning and multiview learning is arguable and even misleading.

Why choose the composite multimodal framework, since there are lots of multimodal learning methods and why the chosen one is widely used?

In the CPM-Net, the versatility also implies the advantages of using multi-view information, and the paper also provides analysis in terms of empirical error.  It will be better if the authors could provide a detailed discussion to clarify the relationship between them and advantages of the proposed one. Providing theoretical results is a good direction but the current results seem a little straightforward.



**Time Spent Reviewing:**

3

---

> ### Author Response · Authors · 2021-08-10
> **Response to Reviewer zJa7**
>
> We thank the reviewer for the insightful and detailed review. We appreciate your constructive and helpful comments.
> #### **Q1**:
>
> Why choose the composite multimodal framework, since there are lots of multimodal learning methods and why the chosen one is widely used?
>
> #### **A1**：
>
> The composite multimodal framework is often observed in applications, e.g., audio-visual speech recognition [2], video classification[6], and action recognition[7], but lacks a formal definition in theory.
>
> Our framework is motivated by the fusion problem, which is one of the most researched aspects of multimodal machine learning with work dating to 25 years ago [6]. In technical terms, it first fuses information from multiple modalities,  which can be regarded as the latent representation mapping $g$,  and then passes the fused result to a task-specific function, to predict an outcome measure of a class through classification, or a continuous value through regression, which corresponds to our task function $h$.  A concrete example of our model about the video classification problem is provided at the end of Section 3 in our paper (line 124-128).
>
> Moreover, for various applications, different fusion techniques:  aggregation-based, alignment-based, Conv,  Channel Exchanging,  Attention [2-7],  can also easily be incorporated into our framework,  by treating the fusion function as a latent space mapping.
>
>
> #### **Q2**:
>
>  It will be better if the authors could provide a detailed discussion to clarify the relationship between  CPM-Net and advantages of the proposed one.
>
> #### **A2**:
>
> We thank the reviewer for suggesting the relevant reference on CPM-Nets [1], which is motivating and insightful. We agree that people sometimes use multi-view and multi-modal interchangeably in machine learning literature. And [1] has a similar purpose to our work, in that the multi-view representation can recover the same performance as only using the single-view observation by constructing the versatility. Its multi-view representation is also related to our latent representation $g$; since they both encode the information from available views/modalities into a latent space.
>
> However, we notice that there are some fundamental differences in our work and [1]:
> 1. Completeness for Multi-View Representation in [1], which requires that each single-view observation can be reconstructed from the latent representation, is not satisfied in our multi-modal setting. For instance, consider the example at the end of Section 3. In a fusion model, the latent representation mapping  $g_{\mathcal{M}}$ has the form  $\varphi_{1} \oplus \varphi_{2} \oplus \cdots \oplus \varphi_{M}$, where $\oplus$ denotes a fusion operation, and $\varphi_{k}$ is a deep network. If we use the sum fusion function, i.e., $g_{\mathcal{M}}= \sum_{k=1}^{M}\varphi_{k}$. Then the inverse mapping of  $g_{\mathcal{M}}$ does not exist, which indicates that only knowing the output of  $g_{\mathcal{M}}$,  one cannot obtain each single-modal observation $\mathbf{x}^{(k)}$.
> 2. The relationships across different modalities are usually uncertain. Hence, it is hard to make strict assumptions on the probability distributions across different modalities. [1] assumed that each view is conditionally independent given the shared multi-view representation. The most related claim to this in the multimodal literature we found is in [8], which stated that conditioning on **ground truth label** $\mathbf{y}$ instead of the shared latent representation, these modalities are conditionally independent. Moreover, as we discussed in Bullet 1, the latent representation is not informative enough to recover every input modality. Therefore, this important conditional independence assumption in CPM-Nets analysis does not hold in our multi-modal setting.
> 3. Although the empirical error is involved in both cases,  the analysis in our work mainly focuses on the generalization side in theory, and does not require additional assumptions on the relationship across every single modality, whereas [1] adapted the standard likelihood analysis from the optimization perspective, which is separated from generalization.
>
> Overall, we agree with the reviewer that [1] is related to our work, and we will include a citation and a discussion comparing this work to ours in the revised paper.
>
> Reference:
>
> [1]Changqing Zhang, Huazhu Fu, Joey Tianyi Zhou, Qinghua Hu, et al. Cpm-nets: Cross partialmulti-view networks. 2019.
>
>
> [2] Tadas Baltrušaitis, Chaitanya Ahuja, and Louis-Philippe Morency. Multimodal machine learning: A survey and taxonomy.IEEE transactions on pattern analysis and machine intelligence,41(2):423–443, 2018
>
> [3] Haytham M Fayek and Anurag Kumar. Large scale audiovisual learning of sounds with weaklylabeled data. 2020
>
> [4] Christoph Feichtenhofer, Axel Pinz, and Andrew Zisserman. Convolutional two-stream networkfusion for video action recognition. InProceedings of the IEEE conference on computer visionand pattern recognition, pages 1933–1941, 2016
>
> [5] Yikai Wang, Wenbing Huang, Fuchun Sun, Tingyang Xu, Yu Rong, and Junzhou Huang. Deep multimodal fusion by channel exchanging. Advances in Neural Information Processing Systems,33, 2020.
>
> [6] Weiyao Wang, Du Tran, and Matt Feiszli. What makes training multi-modal classification networks hard? In 2020 IEEE/CVF Conference on Computer Vision and Pattern Recognition (CVPR), pages 12692–12702. IEEE, 2020.
>
>
> [7] M Esat Kalfaoglu, Sinan Kalkan, and A Aydin Alatan. Late temporal modeling in 3d cnn architectures with bert for action recognition. In European Conference on Computer Vision, 346 pages 731–747. Springer, 2020.
>
> [8] Xinwei Sun, Yilun Xu, Peng Cao, Yuqing Kong, Lingjing Hu, Shanghang Zhang, and Yizhou Wang. Tcgm: An information-theoretic framework for semi-supervised multi-modality learning. In European Conference on Computer Vision, pages 171–188. Springer, 2020.

---

> > ### Comment · Reviewer_zJa7 · 2021-08-30
> > **Thanks for the response - main concerns is not well addressed**
> >
> > I thank the authors for the detailed reply. I have read the comments from other reviewers and responses, I think in current state it is a borderline paper since some concerns still should be addressed. One of my main concerns is that the authors only provide analysis on a composite multimodal learning model, where the impact is narrow for multimodal learning. There are quite a lot of multimodal learning methods that the analysis may be not suitable for. Accordingly, to some extent, the title is a little big and misleading.
> >
> > Moreover, there are also some papers providing analysis about why multimodal learning is better (e.g., analysis on CCA). So, it is not proper to claim “the first theoretical analysis”.

---

> > > ### Author Response · Authors · 2021-08-30
> > > **Response to Reviewer zJa7**
> > >
> > > We thank the reviewer for the feedback.  We address the concerns you raise below.
> > >
> > > #### **Q**.  One of my main concerns is that the authors only provide analysis on a composite multimodal learning model
> > > #### **A**.
> > > As we pointed out in our reply in A1, the composite multimodal framework is often observed in applications, and is also motivated by the fusion problem, which is one of the most researched aspects of multimodal machine learning with work dating to 25 years ago.
> > >
> > > In fact, in recent years, a large number of papers, e.g., [1-6], have utilized this framework in one way or another, although the contributors did not clearly summarize the relationship between their methods and this common underlying structure.
> > >
> > > Below, we explicitly discuss how the methods in [1-6]  fit well into our general model, by providing the corresponding function class $\mathcal{G}$ under each method.
> > >
> > > 1. Audiovisual fusion for sound recognition in [2]: The audio and visual models map the respective input to segment-level representations, which are then used to obtain single-modal predictions, $h_a$ and $h_v$, respectively. The attention fusion function $n_{attn}$, ingests the single- modal predictions, $h_a$ and $h_v$, to produce weights for each modality, $\alpha_{a}$ and $\alpha_{v}$. The single-modal audio and visual predictions, $h_a$ and $h_v$, are mapped to $f_a$and  $f_v$ via functions $n_{a}$ and $n_{v}$ respectively, and fused using the attention weights, $\alpha_{a}$ and $\alpha_{v}$. In summary, $g$ has the form: $g = h_{a v}=\alpha_{a} \odot f_{a}+\left(1-\alpha_{a}\right) \odot f_{v}$
> > >
> > > 2. Channel-Exchanging-Network in [4]:  A feature map will be replaced by that of other modalities at the same position, if its scaling factor is lower than a threshold. $g$ in this problem can be formulated as a multi-dimensional mapping $g:=(f_1,\cdots,f_M)$, where subnetwork $f_m(x)$ adopts the multi-modal data $x$ as input and fuses multimodal information by channel exchanging.
> > >
> > > 3. Methods in [3][5][6]: They can be formulated into the form we mentioned in our reply in A2.  Specifically, recall our reply in A2.1, $g$ has the form: $\varphi_{1} \oplus \varphi_{2} \oplus \cdots \oplus \varphi_{M}$, where $\oplus$ denotes a fusion operation, (e.g., averaging, concatenation, and self-attention), and $\varphi_{k}$ is a deep network which uses each modality data $x^{(k)}$ as input. Under these notations,
> > >
> > >      *a*.For the early-fusion BERT method in [6], the temporal features are concatenated before the BERT layer and only a single BERT module is utilized. Here, the $\oplus$ is a concatenation function, and $g$ has the form $(\varphi_{1} ,\varphi_{2} )$.
> > >
> > >      *b*. [3], [5] discussed different fusion methods by choosing $\oplus$. (i) Max fusion: the $\oplus$ is the maximum function and $g:=\max (\varphi_{1},\cdots, \varphi_{M})$; (ii) Sum fusion:  $g:=\sum \varphi_{m}$; (iii) averaging; (iv) self-attention and so on.
> > > 4. The fusion section in the survey [1] provides many works which can be incorporated into our framework.
> > >
> > >
> > > #### **Q**.  The claim “the first theoretical analysis”.
> > >
> > > #### **A**.
> > > We thank the reviewer for the suggestions. We will modify the statement in our revision to make it more accurate.
> > >
> > >
> > > Reference:
> > >
> > > [1] Tadas Baltrušaitis, Chaitanya Ahuja, and Louis-Philippe Morency. Multimodal machine learning: A survey and taxonomy.IEEE transactions on pattern analysis and machine intelligence,41(2):423–443, 2018
> > >
> > > [2] Haytham M Fayek and Anurag Kumar. Large scale audiovisual learning of sounds with weaklylabeled data. 2020
> > >
> > > [3] Christoph Feichtenhofer, Axel Pinz, and Andrew Zisserman. Convolutional two-stream networkfusion for video action recognition. InProceedings of the IEEE conference on computer visionand pattern recognition, pages 1933–1941, 2016
> > >
> > > [4] Yikai Wang, Wenbing Huang, Fuchun Sun, Tingyang Xu, Yu Rong, and Junzhou Huang. Deep multimodal fusion by channel exchanging. Advances in Neural Information Processing Systems,33, 2020.
> > >
> > > [5] Weiyao Wang, Du Tran, and Matt Feiszli. What makes training multi-modal classification networks hard? In 2020 IEEE/CVF Conference on Computer Vision and Pattern Recognition (CVPR), pages 12692–12702. IEEE, 2020.
> > >
> > > [6] M Esat Kalfaoglu, Sinan Kalkan, and A Aydin Alatan. Late temporal modeling in 3d cnn architectures with bert for action recognition. In European Conference on Computer Vision, 346 pages 731–747. Springer, 2020.

---

### Decision · Program_Chairs · 2021-09-27

**Decision:**

Accept (Poster)

**Comment:**

The paper presents a formal theoretical analysis of the utility of observing multimodal information over unimodal information from the perspective of empirical risk minimisation. The analysis is followed up with some empirical evidence seeking to confirm theoretical findings.

The reviewers all agreed that the theoretical contributions were both sound and principled.

There were a few concerns about related work (eg. CPM-Nets) and making things clear and precise that the authors should address.  For example the characterisation of [35] as 'widely-used' is odd to say the least.

The main concern with this paper is with the experiments---specifically the design and the observed results.

It appears to be the case, as the authors themselves point out (l265-267), that the complexity of the function class has a pretty strong effect on the _significance_ of the results (cf. Table 3). The 'bold' values are only significant when the ratio of sample size is 1.0; in all other cases, they're within the reported standard deviations, and cannot be claimed to particularly significant.

Moreover, as reviewer VZFZ points out with relevant citations, in practice, multimodal learning tends to perform worse. In the discussion, this is somewhat hand-waved away as having to do with optimisation, but it would seem to be a pretty serious issue affecting the theory if it is something typically not observed in practice!
A proper discussion of what the cause for this apparent contradiction is and how it might relate to the complexity of the function class/sample size ratio in such cases would strengthen the paper.

I would also urge the authors to incorporate the clarifications and edits in discussion with all the reviewers, especially reviewer eSGN into the updated manuscript.

If there such a thing as a conditional accept, this paper would be a prime candidate as it does have relevant contributions, but also leaves a few things to be desired before it can be classed a complete piece of publishable work.
It would probably be reasonable to accept the paper on the merits of its theoretical contributions, but *strongly* encourage the authors to address the question of applicability, relation to contradictory results in practice, how this relates to the function-class complexity vs empirical observations trade-off they discuss (esp Sec 5.1).